

# Satellite ice extent, sea surface temperature, and atmospheric methane trends in the Barents and Kara seas

Ira Leifer[1], F. Robert Chen[2], Thomas McClimans[3], Frank Muller Karger[2], Leonid Yurganov[4]

[1] Bubbleology Research International, Inc., Solvang, CA, USA

[2] University of Southern Florida, USA

[3] SINTEF Ocean, Trondheim, Norway

[4] University of Maryland, Baltimore, USA

Correspondence to: Ira Leifer (ira.leifer@bubbleology.com)

**Abstract.** Long-term (2003-2015) satellite-derived sea-ice extent, sea surface temperature (*SST*), and lower tropospheric methane ($CH_4$) of the Barents and Kara Seas (BKS) were analyzed for statistically significant anomalies and trends for 10 focus areas and on a pixel basis that were related to currents and bathymetry. Large positive $CH_4$ anomalies were discovered around Franz Josef Land (FJL) and offshore west Novaya Zemlya in early fall. Far smaller $CH_4$ enhancement was around Svalbard, downstream of known seabed seepage.

Strongest *SST* increase was southeast Barents Sea in June due to strengthening of the warm Murman Current (MC) and in the south Kara Sea in September, when the cold Percey Current weakens and the MC strengthens. These regions and around FJL exhibit the strongest $CH_4$ growth. Likely sources are $CH_4$ seepage from subsea permafrost and hydrates and the petroleum reservoirs underlying the central and east Barents Sea and the Kara Sea. The spatial pattern was poorly related to depth and better explained by shoaling. Peak $CH_4$ anomaly is several months after peak *SST*, consistent with a several month delay between *SST* and seabed temperature. Continued MC strengthening will increase heat transfer to the BKS, rendering the Barents Sea ice-free in about 15 years.

**Keywords:** Arctic, methane, sea surface temperature, ice, Barents Sea, warming, currents, emissions

Highlights:

• Warm, northwards flowing Murman Coastal Current penetrates further into the Barents and Kara seas

• Currents transport positive SST and drive growing methane emissions

• Shoaling provides a mechanism that allows deep-water methane to reach the atmosphere

• Franz Josef Land and the west coast of Novaya Zemlya are important, un-inventoried and growing $CH_4$ sources




## 1. Introduction

### 1.1 Changes in the Arctic Environment in the Anthropocene

Over recent decades, the Arctic has been experiencing amongst the fastest changes from global warming, termed Arctic amplification (Manabe and Stouffer, 1980) with the Arctic Ocean warming at nearly double the rate of the rest of the world's oceans (Hoegh-Guldberg and Bruno, 2010). Arctic amplification is strongly evident in the progressive reduction of sea ice cover over the Arctic Ocean associated with sea surface temperature increases (Comiso, 2012; Comiso et al., 2008; Graversen et al., 2008; Hoegh-Guldberg and Bruno, 2010; Overland and Wang, 2013; Screen and Simmonds, 2010; Stroeve et al., 2014). Data since 1948 show that Arctic Ocean and atmospheric temperatures and storm frequency increased as sea ice extent and volume decreased (NRC, 2014).

One of the most evident, recent manifestations of Arctic change is the observed progressive decrease in sea-ice extent, which underlies numerous ocean physical (NRC, 2014) and ecosystem feedbacks (Alexander et al., 2018). These include heat transfer, light availability in the water column, and momentum transfer (convective and wind mixing), as well as ocean heat and moisture exchange with the atmosphere. Sea ice changes affect weather and overall surface albedo (NRC, 2014).

The Arctic has global impacts by directly affecting weather including extremes at mid-latitudes (Cohen et al., 2014), by affecting global albedo and hence the earth's energy balance (NRC, 2014), and by contributing to the budgets of methane ($CH_4$), a potent greenhouse gas with greater radiative impact on a 20 year time scale than carbon dioxide (IPCC, 2007; Fig. 2.21). Since pre-industrial times, $CH_4$ emissions have risen by a factor of 2.5 (Dlugokencky et al., 2011). Inventories have high uncertainty currently (IPCC, 2013), which is greater in future projections (Saunois et al., 2016), with the Arctic contributing large uncertainty. Underlying the latter are the vast Arctic $CH_4$, $CH_4$ hydrates, and organic material stores trapped under permafrost both onshore (Tarnocai et al., 2009) and offshore (Archer et al., 2009). Rapid warming shallow of Arctic marginal seas will degrade the integrity of submerged permafrost and its sequestered $CH_4$ (Shakhova et al., 2017). As a result, Arctic marine $CH_4$ emissions are being driven by the processes that are underlie this marine warming including retreat of sea ice, heat transport by currents, and increased solar energy inputs that arises in part from the higher albedo associated with sea ice retreat.

FIGURE 1 HERE

Arctic amplification has implications for seabed methane ($CH_4$) emissions – particularly for seabed $CH_4$ currently "sequestered" beneath subsea permafrost – terrestrial permafrost inundated by rising sea level after the Holocene. For example, extensive seabed $CH_4$ seepage is linked closely with destabilization of subsea permafrost in the East Siberian Arctic Sea (Shakhova et al., 2013) and has been estimated as comparable to emissions from Arctic Tundra (Shakhova et al., 2015). Warmer seabed temperatures will degrade subsea permafrost integrity further (Shakhova et al., 2017), enhancing emissions (Shakhova et al., 2015); however, time scales remain uncertain.

The marine Arctic also is affected by changes in the terrestrial Arctic, not just climate, but also fresh water and organic material inputs from rivers. Arctic soils contain 50% of the global subterranean carbon pool of which 88% is estimated sequestered in permafrost (Tarnocai et al., 2009). A quarter of this 1670 Pg (1 Pg=$10^{15}$g) carbon pool may be mobilized into the Arctic Ocean and sub-marginal seas over the next century due to Arctic warming (Gruber et



al., 2004). The Arctic and sub-Arctic show strong terrestrial, high latitude, CH$_4$ anomalies for eastern Canada,
Alaska, and Western Russia (**Fig. 1**). Still, the Barents Sea, where the most rapid ice loss has occurred (Onarheim
and Årthun, 2017) and the Kara Sea, show the strongest anomalies by far.

**1.2. Study Motivation**
We hypothesize that both lower tropospheric CH$_4$ and primary productivity – two parameters that are amenable
to remote sensing – correlate with changes in the overall water column temperature more directly than to changes in
sea surface temperature (*SST*), another satellite remote sensing product. Satellite data are key as they allow repeat
observations of multiple variables on synoptic scales. In this study, we investigate atmospheric CH$_4$ and *SST* trends
to determine the relationship in the spatial distribution of Arctic Ocean tropospheric CH$_4$ with respect to areas that
become ice-free seasonally and inter-annually.
The largest Arctic sea ice cover reductions over the last decade have occurred in the Barents Sea, which likely
will be the first ice-free arctic sea. Sea ice reduction directly affects CH$_4$ flux to the atmosphere by no longer
impeding gas transfer, but also may relate to other oceanographic changes that affect regional ocean-atmospheric
CH$_4$ fluxes. Specifically, we propose that warming *SST* relates to warmer seabed temperatures and hence CH$_4$
emissions from subsea permafrost and hydrate destabilization. The relationship between CH$_4$ seepage and
atmospheric CH$_4$ is indirect – seabed CH$_4$ must be transported across the water by bubbles, diffusion, vertical
mixing, and advection on timescales faster than microbial oxidation timescales.
We investigate this question by analyzing the timing of sea ice retreat and identifying statistically significant
*SST* and atmospheric CH$_4$ trends from satellite data for 2003-2015.
FIGURE 2 HERE
The potential for this approach was revealed in a scoping study of a small area (**Supp. Table S1, Box A2**) in the
marginal ice zone where Barents Sea water flows into St. Anna Trough between Franz Josef Land and Novaya
Zemlya (**Fig. 2b, star**). For these pixels, satellite observations 2003-2015 show a correlation between CH$_4$ and *SST*
for one of two pixel populations (**Fig. 3**). Based on this result, we test our hypothesis in marginal ice zones where
the relationship is predicted to be strongest. Given that the oceanography, including ice, varies dramatically across
the Barents and Kara seas we select ten sub-areas to elucidate how trends differ across these marginal Arctic seas.
Sub-areas were large enough to decrease noise and small enough not to reduce trends from averaging.
FIGURE 3 HERE
**1.3 Global and Arctic Atmospheric Methane**
The atmospheric CH$_4$ concentration has been rising in recent years (Nisbet et al., 2014) and depends on the
balance between sources and sinks – primarily hydroxyl (OH). Several processes may explain this trend including
increasing Arctic emissions, wetland emissions, fossil fuel emissions, and/or decreasing losses from OH (Ghosh et
al., 2015; John et al., 2012; Nisbet et al., 2014; Turner et al., 2016) compensated by decreasing biomass burning
(Desjardins et al., 2018). The current atmospheric lifetime of CH$_4$ is ~8.5-9 yr (Sonnemann and Grygalashvyly,




2014; Voulgarakis et al., 2013). Rigby et al. (2017) suggest a decline in OH likely contributed to $CH_4$ increases
since 2007. Many $CH_4$ source estimates have a large uncertainty (IPCC, 2013; Saunois et al., 2016) with future
emission estimates having still higher uncertainty (Prather et al., 2012).

Uncertainty is particularly acute for Arctic sources. Global atmospheric $CH_4$ concentrations are highest in the

Arctic, increasing poleward (Xiong et al., 2016). Supporting this enhancement are strong Arctic $CH_4$ sources,
including seabed emissions, terrestrial riverine runoff (Shakhova et al., 2013), and atmospheric transport from
terrestrial sources (industrial, permafrost, wetlands, fires, etc.). Given that Arctic OH concentrations are less than at
lower latitudes (Liang et al., 2017), winter Arctic $CH_4$ lifetime is longer than in summer. Arctic OH exhibits a
seasonal variability that imposes an approximately 10 ppb seasonality in $CH_4$ concentrations (Thonat et al., 2017).
Future uncertainty in Arctic sources is magnified by the strong and continued projected Arctic warming (Graversen
et al., 2008).

Arctic seabed $CH_4$ sources include thermogenic (geological) seepage (Shakhova et al., 2013), biogenic $CH_4$

production (James et al., 2016; Reeburgh, 2007) and submerged permafrost, originally from biogenic and/or
thermogenic sources (Shakhova et al., 2013). Seabed emissions largely are bubbles or dissolved gas; however,
microbial oxidation in near seabed sediments (the microbial filter) limits the importance of dissolved seabed $CH_4$
fluxes (Reeburgh, 2007). Bubble seepage directly transports $CH_4$ up into the water column and potentially the sea
surface after losses through dissolution. Bubble seepage also indirectly transports fluid with dissolved $CH_4$ (Leifer
and Patro, 2002). The fate of dissolved seep $CH_4$ depends strongly on its dissolution depth (Leifer and Patro, 2002)
with microbial oxidation expected to remove dissolved $CH_4$ below the Winter Wave Mixed Layer (WWML)
(Rehder et al., 1999), whereas dissolved $CH_4$ in the WWML mostly escapes to the atmosphere. The fraction of seep
$CH_4$ that dissolves below versus within the WWML depends strongly on seabed depth, volume flux (Leifer and
Patro, 2002), plume synergies that include the upwelling flow (Leifer et al., 2009), and bubble surface properties
including the presence of impurities on the bubble surface (Leifer and Patro, 2002). Frequent Arctic storms both
deepen the WWML significantly and efficiently sparge dissolved $CH_4$ to the atmosphere (Shakhova et al., 2013).
Field studies and numerical modeling have documented that even for deepsea seepage (to ~1 km) seep bubbles can
transport some of the $CH_4$ to the upper water column and potentially sea surface due to plume and deepsea bubble
processes (MacDonald, 2011; Rehder et al., 2009; Solomon et al., 2009; Warzinski et al., 2014).

### 1.4. Arctic Sea Surface Temperature

Arctic sea surface warming arises from several factors, including the distributions of cloud cover, sea ice, ocean

color, upper-ocean stratification, and heat transport between the world's oceans. Sea ice significantly reduces
surface albedo in the Arctic summer; thus, its absence increases absorption of solar insolation by Arctic Ocean
waters. This may lead to a feedback wherein reduced Arctic Ocean ice cover drives greater warming and further
decreasing ice. For example, anomalously warm Barents Sea *SST* (>+2° C in 2015 relative to 1982-2010) may be
associated with low sea-ice cover in this region and exposure of surface waters to direct solar heating (Timmermans,
2016). Ocean color affects the vertical profile of absorption of solar energy with near-surface heating more likely
lost to the atmosphere than deeper heating. Visible solar energy is absorbed in near surface waters by both



phytoplankton and other suspended and dissolved organic matter (DOM) that originates in riverine discharges
(Stedmon et al., 2011). The colored fraction of DOM (CDOM) reduces available light across the water column,
suppressing photosynthesis and increasing stratification (Granskog et al., 2007).

Stratification plays an important role in the Barents Sea energy budget. Barents Sea water column structure is

modulated by winter cooling of surface waters and their convective mixing as well as brine ejection of fresh water
during ice formation. Winter vertical mixing extends to the seabed over large portions of the shallow (200-300 m)
Barents Sea. In spring, the warming of surface waters and freshwater from melting ice support water column
stability and stratification in the central and southern Barents Sea (Loeng, 1991). Stratification isolates deeper waters
from the atmosphere and prevents vertical mixing of dissolved $CH_4$, trapping it in deeper water (Leifer et al., 2015)
while also isolating it from heat exchange with the atmosphere. Coastal waters off Norway and the Murman coast
remain stratified year round due to terrestrial freshwater inputs. Offshore, stratification strengthens in the spring as
surface waters warm and ice melts (Loeng, 1991).

Currents drive heat exchange between the Arctic Ocean and the Atlantic and Pacific Oceans. These exchanges

are major drivers of Arctic Ocean spatial thermal heterogeneity with additional inputs in coastal waters from riverine
outflow. Atlantic currents are the major contributor of oceanic heat to the Arctic climate system on annual (Lien et
al., 2013) and seasonal time scales (Lien et al., 2017). These currents include the West Spitsbergen Current
(Piechura and Walczowski, 2009), the Bear Island Channel Current (BICC), the Murman Current (MC), and the
Norwegian Atlantic Current (NAC) (see **Fig. 4a**), which transport Atlantic Ocean heat to the Arctic through the
Barents Sea. The south fork of the NAC is entrained into the Norwegian Coastal Current (NCC), which is 90%
Atlantic water and 10% river discharge (Skagseth et al., 2008). Variability appears related to the North Atlantic
Oscillation (NAO) with higher Barents Sea temperatures during the NAO's positive phase (Dickson et al., 2000).
This transport has caused significant warming and ice retreat in this area of the Arctic Ocean (Smedsrud et al.,
2013). Winds modulate the volume flow of Atlantic water into the Barents Sea – stronger in winter and weaker is
summer (Stiansen et al., 2009; Fig, 2.3.4). Ice processes further complicate the re-distribution of heat for surface
Arctic Ocean waters – ice coverage insulates the water from atmospheric cooling (heat transfer to the atmosphere),
better preserving its heat and thereby furthering heat transport into the Arctic (Lien et al., 2017; Lien et al., 2013).
Sea temperature lags atmospheric temperatures by 2-3 months, peaking for the Kola area (offshore Murman, Russia)
between 0 and 200 m in September-October, whereas air temperature peaks in July (Stiansen et al., 2009, Figs.

2.3.3, 2.3.8).

Positive feedbacks underlie Arctic amplification. For example, sea-ice reduction increases albedo and greater

heat absorption in upper water layers (IPCC, 2013). There also are more complicated feedbacks. The progression of
warmer water into the Barents Sea drives local winds that decrease wind-advection of sea ice, with decreased sea ice
coverage *increasing* heat loss from the atmosphere (Lien et al., 2017).
**1.5. Airborne and Satellite observations of Tropospheric Methane**

Although the Arctic covers a vast territory, our knowledge of Arctic processes is highly limited both in spatial

coverage and seasonal coverage. This is due to high cost, logistical challenges, and the harshness of Arctic weather.





Satellite sensors have advantages for Arctic observations including quick revisit times and synoptic coverage (Leifer
et al., 2012) and can fill the significant existing temporal and spatial gaps between the few airborne field datasets.
*1.5.1. Airborne Measurements*
A few airborne campaigns have been conducted to measure Arctic atmospheric $CH_4$ since 2005. Given the
highly extensive spatial scales of the Arctic, these campaigns provide only a few summer snapshots of a highly
variable domain.
$CH_4$ concentration profiles over the Arctic Ocean were measured on five flights during the HIAPER Pole-to-
Pole Observations (HIPPO) campaign (Kort et al., 2012; Wofsy, 2011) and produced evidence of sea surface $CH_4$
emissions from the northern Chukchi and Beaufort Seas in most profiles, up to 82°N. Enhanced concentrations near
the surface of the Ocean were common over fractured floating ice in sample profiles collected on 2 Nov. 2009, 21
Nov. 2009, and 15 Apr. 2010. On 13 Jan. 2009 and 26 Mar. 2010, when the seasonally highest level of sea-ice
coverage occurred, $CH_4$ emissions were weak or non-existent. Some of the observational variability was correlated
with carbon monoxide (CO), indicating terrestrial origin.
The Carbon in Arctic Reservoirs Vulnerability Experiment (CARVE) program sought to quantify Alaskan $CO_2$
and $CH_4$ fluxes between the atmosphere and surface terrestrial ecosystems. Intensive aircraft campaigns with
ground-based observations were conducted during summer from 2012-2015 (Chang et al., 2014). No open ocean
measurements were made. Additional Alaskan airborne data were collected summer 2015 (Jun.-Sept.) by the
Atmospheric Radiation Measurements V on the North Slope of Alaska (ARM-ACME) project (38 flights, 140
science flight hours), with vertical profile spirals from 150 m to 3 km over Prudhoe Bay, Oliktok Point, Barrow,
Atqasuk, Ivotuk, and Toolik Lake. Continuous data on $CO_2$, $CH_4$, CO, and nitrous oxide, $N_2O$, were collected
(Biraud, 2016).
West of Svalbard, an area of known widespread seabed $CH_4$ seepage aligned along a north-south fault parallel
to the coast (Mau et al., 2017; Westbrook et al., 2008) was the focus of a field airborne campaign June–July 2014
(Myhre et al., 2016). Flights were conducted using the Facility for Airborne Atmospheric Measurements (FAAM) of
the Natural Environment Research Council (NERC, UK). The campaign measured a suite of atmospheric trace gases
and was coordinated with oceanographic observations. Seabed $CH_4$ seepage led to significantly increased seawater
$CH_4$ concentrations. However, no significant atmospheric $CH_4$ enhancement was observed for the region above the
seeps for data collected summer, 20 Jun. – 1 Aug. 2014 (Myhre et al., 2016), albeit under a period of mostly light
winds.
*1.5.2. Satellite*
Satellite observations provide long-term temporal context for campaign data, which are necessarily, limited in
time, often limited in spatial coverage, and often only occur during a specific season when weather is acceptable for
flight logistics. As such, satellite observations complement high spatial-resolution airborne and boat-based field
observations, which are infrequent and sparse at best. Remote sensing measures column gas abundance and thus is
independent of potential mismatches between the platform altitude and the altitude of enhanced $CH_4$. Airplanes may



not fly sufficiently low to collect data in the planetary boundary layer (PBL), which often is shallow in the marine
Arctic (Aliabadi et al., 2016).
Satellite sensors leverage $CH_4$ spectral features at 1.67 and 2.32 μm in the Short Wave InfraRed (SWIR) (Clark
et al., 2009) and around 7.28 μm in the thermal infrared (TIR) (Tratt et al., 2014). $CH_4$ retrievals for SWIR sensors,
which use passive reflective solar radiance, are challenged in the Arctic by high cloud cover, low solar zenith angle,
and low reflectivity for ice, snow, and water (Leifer et al. 2013). TIR sensors measure emissivity radiance, and thus
comparatively shorter path lengths at high latitudes relative to SWIR sensors that measure reflected sunlight. Also,
TIR sensors can retrieve $CH_4$ above low clouds, both daytime and nighttime – SWIR is only daytime and requires
cloud free skies. Thus, TIR sensors have significant Arctic advantages for marine $CH_4$ retrievals compared to SWIR
sensors (Leifer et al. 2013). However, whereas SWIR sensors primarily respond to near-surface $CH_4$, TIR retrievals
generally have higher sensitivity to mid-tropospheric $CH_4$ than to near-surface $CH_4$ (Xiong et al., 2013).
Recent SWIR satellite $CH_4$ sensors include the recent Scanning Imaging Absorption SpectroMeter for
Atmospheric CHartographY (SCIAMACHY-ESA: 2002-2012: 100-km resolution) mapping mission (Bovensmann
et al., 1999) and the active, Greenhouse Orbiting Satellite (GOSAT-JAXA: 2009-: 9-km resolution) sampling
mission (Kuze et al., 2009). The TROPOspheric Monitoring Instrument (TROPOMI-ESA: 2017– at 7-km
resolution) mapping mission (Veefkind et al., 2012) returned first images in 2017. The scheduled geostationary
CARBon cycle observatory (GEOCARB-NASA; at 4-km resolution) mapping mission will return hourly (daytime)
revisit data for North America (Rayner et al., 2014) in the early 2020s.
In the TIR, the AIRS (Atmospheric InfraRed Sounder) mission onboard the Earth Observation Satellite, Aqua
satellite (Aumann et al., 2003b) and the EuMetsat IASI-1 (InfraRed Atmospheric Sounder Interferometer) mission,
on the MetOp-A platform (Clerbaux et al., 2009) provide long-term $CH_4$ observations. Accompanying IASI-1
(2007-) is IASI-2 (2013-) on the currently orbiting MetOp-B meteorological satellite. The IASI satellites follow sun
synchronous orbits. Additionally, three IASI New Generation instruments (Crevoisier et al., 2014) are planned for
launch in 2021, 2028, and 2035 (IASI-NG, 2017). AIRS $CH_4$ profiles are retrieved from the 7.8 μm TIR channel
(Aumann et al., 2003b).
The IASI instruments are cross-track-scanning Michelson interferometers that measure in 8461 channels at 0.5
$cm^{-1}$ spectral resolution from three spectrometers spanning 645 to 2760 $cm^{-1}$. These spectrometers have a 2×2 array
of circular footprints with a nadir spatial resolution of 12 km that is 39 × 25 km at swath (2400 km) maximum
(Clerbaux et al., 2009). IASI-1 was launched into an 817 km-altitude polar orbit on 19 Oct. 2006, while IASI-2 was
launched on 17 Sept. 2012. MetOp-A and MetOp-B cross the equator at approximately 09:30 and 21:30 local time,
separated by approximately half an orbit, resulting in twice daily, near-global coverage with 29 day. The on-flight
noise-equivalent delta temperature at 280K is estimated to be well below 0.1K in the spectral range of interest to
$CH_4$ (Razavi et al., 2009). Like AIRS, IASI has a wide swath with a scan angle of ±48.3°. IASI $CH_4$ retrieval
algorithms are described by Xiong et al. (2013) and Gambacorta (2013).
AIRS is a grating diffraction nadir cross-track scanning spectrometer on the Aqua satellite (2002-) that is part of
the Earth Observation System (Aumann et al., 2003a). AIRS was launched into a 705-km-altitude polar orbit on the
EOS Aqua spacecraft on 4 May 2002. The satellite crosses the equator at approximately 01:30 and 13:30 local time,





producing near global coverage twice a day. Effective field of view after cloud clearing, as described by Susskind et
al. (2006), is 45 km and the spectral resolution for $CH_4$ is 1.5 cm$^{-1}$. Version 6 of AIRS Levels 2 and 3 data are
publicly available (AIRS, 2016) – see Xiong et al. (2010) for a description, evaluation, and validation of global $CH_4$
AIRS retrievals. Lower-troposphere (0-4 km altitude averaged) AIRS profiles (AIRS time series is longer than IASI)
are analyzed herein.

Validation is critical to any remote sensing approach and has been addressed in a number of studies for the

lower and mid-upper Arctic troposphere. Xiong et al. (2010) compared aircraft data taken over Poker Flat,
Alaska, and Surgut, Siberia with AIRS $CH_4$ retrieved profiles and found agreement within 1.2% with mean
measured $CH_4$ concentration between 300–500 hPa; correlation coefficients were ~0.6-0.7. A significantly wider
geographical coverage was achieved for IASI validation (Xiong et al., 2013) during a quasi pole-to-pole flight of the
National Science Foundation's Gulfstream V aircraft (Wofsy, 2011). A bias of nearly -1.74% was found for 374–
477 hPa and -0.69% for 596–753 hPa. Yurganov et al. (2016) compared 5-year long IASI data for 0-4 km layer over
a sea area adjacent to the Zeppelin Observatory, Svalbard, Norway, at 474 m altitude, operated by the Norwegian
Institute for Air Research (NILU). Monthly mean values and monthly trends were in good agreement, but daily
excursions did not correlate. Yurganov et al. (2016) explained the latter by the observatory's location being near the
top of the planetary boundary layer.

IASI has been used to study lower (<4 km) tropospheric $CH_4$. Yurganov et al. (2016) found low atmospheric

$CH_4$ anomalies in summer for 2010-2015 with annual Arctic Ocean $CH_4$ fluxes estimated as being ~2/3 those from
the terrestrial Arctic. Positive $CH_4$ anomalies were observed along the coasts of Norway, Novaya Zemlya, and the
Svalbard archipelago primarily during November-January (Yurganov and Leifer, 2016a). A breakdown of the Arctic
oceanic summer thermal stratification by wind-induced mixing in autumn may underlie this seasonal trend. A
breakdown of stratification is proposed for $CH_4$ emissions to the atmosphere from the North Sea, which also is
highly stratified in the summer and fall (Leifer et al., 2015). Additionally, Yurganov and Leifer (2016b) report
significant $CH_4$ increases during the 2015/2016 winter over the Sea of Okhotsk.
**2. Method and Study Design**
**2.1. Overview**

In this study, we characterize several processes by satellite observations aggregated on a monthly basis. Satellite

data allow repeat regional observations spanning many years. Specifically, we investigated the relationship between
ice-free months, sea surface temperature (*SST*), and the atmospheric $CH_4$ column. We concentrate on five area types:
(1) Arctic water affected areas; (2) combined Arctic and Norwegian Atlantic Current affected areas; (3) Barents Sea
Polar Front affected areas; (4) Murman Current affected areas; and (5) Norwegian Coastal Current and Murman
Coastal Current affected areas.

Specifically, satellite products for the Barents and Kara Seas are quality reviewed and then analyzed to identify

statistically significant trends on both a pixel basis and in focus areas relative to regional trends (**Section 2.2**). The
analysis uses relative trends to reduce potential retrieval biases and uncertainty. The use of focus areas allows pixel





aggregation to reduce the impact of a highly spatially heterogeneous signal and to reduce the effect of inter-annual
spatial shifts, which could appear as local temporal variations.
The analysis then investigates these trends in relationship to oceanographic and meteorological processes and
available, Barents and Kara Seas data relevant to heat transport to, within, and between the Barents and Kara Seas
(**Section 2.3**). This analysis investigates the importance of different processes to improve our understanding of the
fate of seabed methane emissions.
FIGURE 4 HERE
**2.2. Methodology**
*Satellite data*
AIRS $CH_4$ data (version 6) are publicly available from NASA Goddard Space Flight Center (GSFC) since 2002
(AIRS Science Team/Joao Texeira, 2016). $CH_4$ data for 2003–2015 are retrieved by the NOAA Unique Combined
Atmospheric Processing System (NUCAPS) algorithm, developed at NOAA/NESDIS in cooperation with Goddard
Space Flight Center (GSFC). Data are analyzed for open ocean areas with high vertical thermal contrast, defined
here as the temperature difference between the surface and altitude of 4 km (Yurganov and Leifer, 2016a; Yurganov
et al., 2016). $CH_4$ data are re-projected to a 4-km azimuthal equal area projection. The $CH_4$ anomaly ($CH_4'$) is
calculated by subtraction of the values computed within each of the 10 focus areas from the average of the whole
Barents Sea ocean climatology. As $CH_4$ shows high inter-annual variability, a three-year running average is applied.
Ocean *SST* are from the Moderate Resolution Imaging Spectroradiometer (MODIS) sensor on the Aqua satellite
(NASA, 2015), obtained from the GSFC, Ocean Ecology Laboratory, Ocean Biology Processing Group (OEL-
OBPG). The 4-km, Level 3 data are re-projected to a 4-km, equal azimuthal area projection. Satellite data products
are cloud screened (Ackerman et al., 2010; Ackerman et al., 1998). The mapped products match the $CH_4$ data
projection.
First, data are quality reviewed for sea ice coverage and cloud coverage filtered for coastlines, which are from
the Global Self-consistent, Hierarchical, High-resolution Shoreline database (SEADAS, 2017). Shape files of sea-ice
monthly extent are obtained from National Snow and Ice Data Center (Fetterer et al., 2017) and are based on
monthly passive microwave radiometry with the Bootstrap algorithm (Comiso et al., 2008). Sea-ice fields are
provided on a polar stereographic grid at 25-km resolution. The number of ice-free months is derived from the
intersection of the monthly ice shape file for each year with the focus areas. The number of ice-free months each
year is tallied by the following rules: if the intersection is less than 15%, it is counted as 0 months; if coverage is
greater than 15% and less than 50% of the pixel, it is counted as 0.5 months. When coverage is greater than 50% in a
single month the pixel is counted as ice covered for the month. Ice-covered (>50%) pixels are not used in the trend
or the climatology calculations for *SST* ($CH_4$ retrievals are accurate over both ice and seawater).
*Trend analysis*
To estimate trends in the Barents Sea and adjacent areas, the monthly mean time series for each grid point in the
images covering this region are calculated. Then, a first order polynomial is calculated by a linear regression





analysis. These linear trends are analyzed using the Mann Kendall Test (Önöz and Bayazit, 2003) and Sen's linear
trend analysis (Juahir et al., 2010; Sen, 1968).

Visual analysis of the trends and anomaly maps of the Barents Sea were used to determine the location of 10

focus areas. The average trends of the 10 focus boxes are calculated from the average of all valid pixels in each
focus box each year for the same months.
**2.3. Setting**
*Oceanography and Meteorology of the Barents and Kara Seas*

Significant Arctic Ocean water derives from the North Atlantic, which becomes denser through cooling.

Additional contributions are riverine and precipitation (which decrease water density) and Pacific water from the
Bering Strait. Most of this water returns to the North Atlantic as part of the global thermohaline circulation (Aagaard
and Carmack, 1989; Carmack and McLaughlin, 2011; Yamamoto-Kawai et al., 2008).

The relatively shallow (230-m average depth) Barents Sea (**Fig. 4**) is characterized by a deep Arctic shelf with

complex bathymetry and hydrography (Loeng, 1991). The Barents Sea is bounded to the south by the northern coast
of Europe, to the north by two archipelagos, Svalbard and Franz Josef Land (FJL), to the east by the large north-
south oriented island, Novaya Zemlya, and to the west by the Norwegian Sea. In winter, the Barents Sea is partially
ice-covered, while it is almost ice-free in the summer (**Fig. 4b**).

Barents Sea physical oceanography is influenced strongly by inflows from the North Atlantic and the Arctic

Ocean. North Atlantic water inflows to the Norwegian Sea, forming the Norwegian Atlantic Current (NAC), one
track of which becomes the West Spitsbergen Current (WSC) in the Greenland Sea and the Fram Strait, before
bearing eastwards to the north of Svalbard into the Arctic Ocean. The remainder flows into the Barents Sea through
the Barents Sea Opening. Whitehead and Salzig (2001) suggested (and demonstrated in the laboratory) that remote
forcing of the NAC through the Barents Sea lifts the current by several hundred meters to the sill of the Bear Island
Channel, forcing significant anticyclonic vorticity. This drives the retrograde Bear Island Channel Current (BICC)
northeast along the slope of Svalbardbanken and the prograde MC along the slope of Tromsøflaket, eastward and
north to the east of Central Bank (Li and McClimans, 1998; Loeng, 1991). These meet east of Central and Grand
Banks. The resulting flow cools from contact with the atmosphere into a denser, modified Atlantic Water flow that
exits through the St. Anna Trough to the east of Franz Joseph Land (Gammelsrød et al., 2009). Cooling at the banks
also produces a dense westward underflow, depicted by the dashed line in Fig. 4a.

The Norwegian Coastal Current (NCC) follows the Norwegian and Murman coastlines and incorporates fresh

water runoff from northern Europe and Atlantic Water. The NCC is a major contributor of oceanic heat to much of
the southern and eastern Barents Sea and into the Kara Sea (Lien et al., 2013). The NCC cools significantly through
interaction with the atmosphere, whereas the western fork of the NAC (the WSC) submerges north of Svalbard
(location varying seasonally) under an isolating layer of colder and fresher water, better preserving its heat (Lien et
al., 2013). The name of the NCC changes to the Murman Coastal Current (MCC) as crosses into Russian waters.
Long-term (1905-) temperature data for the upper 200 m are available from a section off the Kola Peninsula (**Fig.**
**4a, KS, black dashed line**), which the MCC crosses (Boitsov et al., 2012). These data reveal long-term trends with





a cooler period from 1875-1930 and continuous warming of ~0.8°C since a minimum in 1970-1980 (Skagseth et al.,
2008). Skagseth et al. (2008) found good agreement in the Kola Section temperature trend with the Atlantic Multi-
decadal Oscillation (AMO) index.
The MCC continues eastward at the border of the White Sea until the western shores of Novaya Zemlya divert
the flow northward. It then continues into the Arctic Ocean through the St. Anna Trough between Franz Josef Land
and Novaya Zemlya (Loeng, 1991), which is the dominant outflow of the Barents Sea (Maslowski et al., 2004). A
fork of the MCC flows eastward into the Kara Sea through narrow and very shallow (20-50 m) straits (see **Supp.**
**Fig. S2** for details of Kara Sea currents).
The Percey Current (PC) transports cold, low saline, Arctic surface water into the Barents Sea east of Svalbard
(**Supp. Fig. S1**). The intersection of the Percey Current with the warmer, high saline waters of Atlantic origin in the
Barents Sea gives rise to the Barents Sea Polar Front (Oziel et al., 2016), whose location is controlled by seabed
bathymetry, i.e., it is semi-stationary (Gawarkiewicz and Plueddemann, 1995). This front is part of a unique frontal
system due to its combination with the seasonally ice-covered zones in the northern, central, and eastern Barents Sea
(Vinje and Kvambekk, 1991). As a result, this front tends to exhibit enhanced phytoplankton blooms (Fer and
Drinkwater, 2014) and variability (Falk-Petersen et al., 2000). The PC merges with the East Spitsbergen Current
(ESC) and flows up the west Spitsbergen coast, inshore of the WSC, as the Spitsbergen Coastal Current (SCC),
leading to the Barents Front looping around Spitsbergen (Svendsen et al., 2002).
Winds in the eastern Barents Sea generally circulate counterclockwise (cyclonically), from the north along
Novaya Zemlya in winter and spring and weakly from the south in summer and fall (Gammelsrød et al., 2009). This
leads to calm over the Central Bank, easterlies to the north of Franz Josef Land and from the north to the west of
Svalbard (Kolstad, 2008; Moore, 2013). The Barents Sea is stormy – over 125 days per year have winds above 15 m
$s^{-1}$, which are mostly from the south (Kolstad, 2008). In spring, winds are similar, but displaced to the south. In the
summer, winds blow to the south over most of the Barents Sea, except from the west off Svalbard, with average
summer winds of ~6 m $s^{-1}$ (Kolstad, 2008). In the fall, winds are similar to the summer, but stronger (~8-10 m $s^{-1}$)
and strongly northerly between Svalbard and Greenland. Decadal-averaged air temperatures on Bear Island have
been rising (~2.5°C) since 1980 (Boitsov et al., 2012), about four times larger than the global atmospheric trend over
the same period of ~0.6°C (http://eca.knmi.nl/).
Kara Sea hydrography is controlled by the freshwater outflow of the Ob and Yenisei Rivers (**Fig. 2b; Supp.**
**Fig. S2**), which contribute 350 and 650 km³ yr⁻¹, respectively (Stedmon et al., 2011), about double that of the
Mississippi. Sediment from these estuaries lead to the northeast Kara Sea being mostly shallow (< 50 m). The
western Kara Sea is deep (mostly >100 m), descending to below 500 m in the Novaya Zemlya Trough (Polyak et al.,
2002). River outflows are primarily (>75%) between May and September. As a result, east Kara Sea surface waters
can be brackish. The inflow of modified Atlantic water from the Barents Sea supplies the deeper water in the trough.
Cooler surface water from the MCC, local runoff, and ice from the north flow into the NZCC, some of which returns
to the Barents Sea through the Kara Strait. The trough is dense and fully saline. Cooler surface water from the Arctic
Ocean flows south in the narrow and weak Novaya Zemlya Coastal Current (NZCC) and exits through the Kara
Straits. Additionally, there is a strong, submerged, cool southward flow of Arctic Ocean water along the Novaya





Zemlya Trough. Warmer water enters from the Barents Sea flows east through the Kara Strait and joins a slope
current to the north. Much of this water mixes with the southern and returns to the Barents Sea through the northern
Kara Strait (McClimans et al., 1999; McClimans et al., 2000). Overall, currents through the Kara Sea are largely
northwards, driven by river outflow. The remainder of the north current flow splits to the west and joins the cold
southward NZCC. Prevailing winds are mostly from the southwest for the western Kara Sea and from the south to
southwest in the central Kara Sea (Kubryakov et al., 2016).
Ten focus areas (**Fig. 4a; Table 1** for locations) were selected based on the location of Barents and Kara Seas
currents and ice formation dynamics to investigate the effect of the significant differences between winter and
summer ice coverage. These focus areas are grouped into 5 oceanographic types. The most northerly focus areas
(A1-A3) characterize the inflow of Arctic surface water through both gaps between the archipelagos of Svalbard and
Franz Josef Land and between Franz Josef Land and Novaya Zemlya. Each of these focus areas exhibits different
seasonal ice coverage. Another group of focus areas are west of Svalbard (A4-A6) and are influenced by the West
Spitsbergen Current and water from the Barents Sea. These areas also are affected by Arctic Ocean ice outflow
along Greenland. A focus area near Bear Island (A7) is affected by the warm, north flowing NAC and the cold,
southwest-flowing Percey Current and is located in a subduction area. Two focus areas (A8 and A10) were selected
that are influenced by the Murman Current and Arctic water from the Percey Current in the Barents Sea Polar Front
region (Harris et al., 1998). Finally, one focus area (A9) is situated in the coastal waters to the west of Novaya
Zemlya and is influenced by the MCC with strongly, seasonally varying ice coverage.
**3.0 Results**
**3.1. Barents Sea In situ**
In situ $CH_4$ measurements were made by cavity enhanced absorption spectroscopy (Los Gatos Research Inc.,
Mountainview, CA). Both transits followed a very similar trajectory (**Supp. Fig. S3**) and found very strong,
localized, $CH_4$ anomalies - see **Supp. Fig. S3** for full dataset. These anomalies were far from shore, indicative of
local (i.e., marine) not distant (i.e., terrestrial) sources. The only reasonable explanation is seep bubble plumes –
vessel exhaust was ruled out - see **Supp. Material Sect. 2** for more details.
FIGURE 5 HERE
There was an abrupt drop in $CH_4$ around 72°N for the outwards transit, which increased again around 75°N.
This depressed $CH_4$ portion of the transit corresponded fairly closely with where the vessel left the warm Murman
Coastal Current (**Supp. Fig. S4a**). The strongest anomaly, to 2000 ppb, was observed on the southwards transit
where the MCC rises over the sill into the Santa Anna Trough (78.7°N), close to the focus area shown in **Fig. 3**.
The two transits were separated by about a month with the September transit higher by ~30 ppb than in August,
consistent with strong seasonal $CH_4$ trends. There were other significant differences. Whereas there were many
narrow, implying local, $CH_4$ anomalies during the southwards transit, there were far more than during the
northwards transit and the peak at 78.7°N was not repeated, indicating variability in the emissions.



The difference between these transits highlights the challenges of interpreting such snapshot data, supported by
the comparison with IASI pixels that were proximal and within several days (**Supp. Fig. S4**). Agreement for the
northwards transit was reasonably good (generally within 10 ppb), and generally poor for the southwards transit.
**3.2. Focused Study Area Annual Trends**
Trends in aggregated pixel "focus areas" are compared for three Barents Sea regions, "Northwest of Barents"
including the Greenland Sea and Fram Strait, west of Svalbard (A4-A6), "Northern Barents" in the marginal ice
zone at the edge of the Arctic Ocean (A1-A3) and "Southern Barents," which is strongly under the heat influence of
the east fork of the NAC (A7-A10). Of these, A7, A8, and A9 also cover banks – offshore areas of elevated seabed
topography. Grouping of focus areas with similar trend patterns into these three Barents Sea regions was based both
on physical oceanography and the detailed character of these trends, described below.
FIGURE 6 HERE
Focus areas with the strongest decreasing ice cover trends are in the marginal ice zone of the northern Barents
Sea (south and east of Franz Josef Land), at the southern margin of the Arctic Ocean (**Fig. 6a**, **A1-A3**). Trends for
these three study areas are very similar (after classifying 2006 and 2014 for focus area A3 (Svalbard Northwest) as
outliers. Note, focus areas A1-A3 show below-trend ice-free months in 2014 despite no significant 2014 *SST*
deviation, supporting its classification as an outlier (**Fig. 7a**).
The similarity in ice coverage trends for area A3 (along the cold Percey Current) with areas A1 and A2 (along
the Murman Current's warm, northward leg) suggests not only increasing northward heat transfer, but also
weakening southward cold-water advection. Area A4 (northwest of Svalbard) also shows decreasing ice coverage
towards more frequent year-round ice-free status and lies at the Arctic Ocean boundary (**Fig. 6b**), albeit more under
the influence of warmer NAC waters than those under the influence of the Murman Current in the north-central
Barents Sea (A1-A3). The Central Bank of the Barents Sea (**Fig. 6c**, **A10**) last saw an ice-covered month in 2005,
while a noisy trend of decreasing ice coverage is evident offshore coastal southwest Novaya Zemlya (**Fig. 6c**, **B9**),
along the western fork of the Murman Coastal Current.
Overall, all focus areas are trending towards year-round ice-free, with the entire Barents Sea likely to be year
round ice free by ~2030 based on an extrapolation of trends in Northern Barents Sea focus areas, A1-A3.
FIGURE 7 HERE
*SST* increases in all focus areas (**Fig. 7**) albeit at rates spanning a wide range, from 0.0018 to 0.15 °C yr$^{-1}$ (see
**Table 1**). In the Northern Barents Sea, the strongest warming trend is for area A1, east of Franz Josef Land. This is
located in a marginal ice zone, in the path of the warm MC. Area A3 shows the weakest warming trend lies along
the cold Percey Current. These trends also are consistent with a strengthening of warm currents and weakening of
cold currents inferred from the changes in ice coverage. For the Northwest of Barents focus areas (**Fig. 7b**, **A4-A6**),
the strongest warming is at the northernmost focus area, A4, whereas the weakest trend is for the southernmost focus
area (**Fig. 7**, **A6**). This also is consistent with a strengthened northwards penetration of the warm NAC forming the
Bear Island Channel Current (BICC).



The strongest warming trend occurs southwest of Novaya Zemlya (**Fig. 7c**, **A9**) along the path of the northerly
turn of the MCC, in shallow water. This trend is consistent with increased eastward MCC penetration east along the
coast of Novaya Zemlya and into the Kara Sea. A very weak and highly variable *SST* warming trend is observed to
the south of the Svalbard Bank at the intersection of the cold Percey Current and the warm BIC (A7). Areas A10 and
A8, and to a lesser extent A9 all show a strong oscillation of ~8 years with peak values in 2005 – 2007, and a
minimum around 2010. The same pattern also is observed to the east of Franz Josef Land (areas A1 and A2). All the
boxes that exhibit this oscillation lie along the Murman Current, whose origin is in the NAC.
FIGURE 8 HERE
A positive $CH_4$ trend is observed across the Barents and Kara Seas from June through September with some
regions exhibiting far stronger trends than average (**Supp. Fig. S5**). Areas of faster $CH_4$ increase include near Franz
Josef Land (**Fig. 8a**, **A1**, **A2**), the shallower waters offshore W. Svalbard (**Fig. 8b**, **A4**), and offshore Novaya
Zemlya (**Fig. 8c**, **A9**). These areas of increasing $CH_4$ correspond to areas of consistent warming for 2003-2015 (**Fig.**
**7a, A1, A2**) and consistent warming since ~2004/2005 for southwest offshore Novaya Zemlya and the Central Bank
of the Barents Sea (**Fig. 7c**, **A8-A10**). All these focus areas lie along the northwards flow of the Murman Current
and the Murman Coastal Current. In contrast, focus areas along the Percey Current show a slowly decreasing $CH_4$
relative to the trend for the entire Barents Sea (**Fig. 8, A3, A7**), despite an (albeit weakly) increasing *SST*.
The strongest $CH_4$ growth is south of Franz Josef Land (**Table 1 A2**, 3.49 ppb yr$^{-1}$), followed by offshore
northwest Svalbard (**Table 1 A4**, 3.37 ppb yr$^{-1}$- 2003-2015, 3.6 ppb yr$^{-1}$ 2005-2015). This positive trend is sustained
over the analysis period. This area off the Fram Strait has natural $CH_4$ seepage associated with hydrate
destabilization (Westbrook et al., 2008). This increase is annualized, and thus does not result from shifts in the
timing of seasonal warming. Note, the $CH_4$ slopes for areas A4-A10 all are larger when calculated from the 2005
minimum, but not for A1-A3 (**Table 1**). The former lie along the NAC and its eastern current fork, the Murman
Current. Neither the Percey Current focus areas (A3, A7) nor other northern Barents Sea focus areas (A1, A2) show
this effect depending on the reference time.
TABLE 1 HERE
The largest variability in *SST* and $CH_4$ was in the focus area to the north of Murman in the MC (**Table 1, A8**;
**Fig. 8**) and likely arises from variations in the strength and location of the MC, which bifurcates around this focus
area. Importantly, this area is located above a small, unnamed bank to the south of the Central Bank (**Fig. 2b**).
FIGURE 9 HERE
**3.3. Barents and Kara Sea Climatology**
Atlantic heat input is very important to the energy budget of the Arctic Ocean and Barents Sea and is driven by
the two forks of the NAC (**Fig. 4a**) (Lien et al., 2013). This importance is evident in the Barents Sea *SST*
climatology (**Fig. 9**). Warmer water flows eastward along the northern Norwegian and Murman coasts and along the
eastern coastline of Svalbard (BIC), and north along the western Svalbard coast. Water becomes cooler as it
penetrates eastward, and as it reaches the ice edge. Across much of the Barents Sea there is a strong latitudinal *SST*
gradient extending south from the ice edge, independent of the location of the eastern NAC branches. In the coastal



waters of Novaya Zemlya, warmer water extends further north than elsewhere. The warm signature disappears in the
area where the NAC submerges, near the northern shores of Novaya Zemlya (**Fig. 4a**).
FIGURE 10 HERE
In June, the warm water extent corresponds fairly well with the median ice edge location, which trends along
the location of the cold Percey Current. Southeast of Svalbard, the Percy Current penetrates southward as a narrow
extension of cold water ending south of Bear Island. Slightly cooler water is observed over the two banks in the
central Barents Sea.
The shift to summer *SST* patterns occurs in July, increasing in August, and then beginning to decrease in
September (**Supp. Fig. S5**). For north Spitsbergen in the Svalbard archipelago (**Supp. Fig. S1**) the northerly cold
Spitsbergen Coastal Current (SCC) inshore of the West Spitsbergen current (WSC) breaks down. This suggests the
SCC is entrained by the more energetic WSC (McClimans, 1994), flowing northwards along southwest Spitsbergen
underneath colder surface waters, likely below strong summer stratification. The WSC warm water is further
offshore the west coast of Svalbard in June than in September (**Supp. Fig. S1**), i.e., the Barents Front shifts
shoreward in summer.
By September, *SST* in the shallower eastern (coastal) Barents Sea has warmed to levels comparable to the
warmer waters in the southwest Barents Sea where NAC heat input maintains elevated *SST*. Warmer *SST* also
extends further offshore Norway and Murman. These seasonal *SST* changes match the sea ice's northwards retreat to
Franz Josef Land (**Fig. 10b**). However, Barents Sea warming does not follow the ice edge between Svalbard and
Franz Josef Land, corresponding instead to the location of the Percy Current. Still, whereas warm water is more
extensive in south and east Barents Sea in September than August, in the northwest, cold water associated with the
Percey Current has expanded from August to September (**Supp. Fig. S5**).
The now mostly ice-free Kara Sea in September exhibits coastal warming, particularly to the east, where there
also is heat input from the Ob and Yenisei Rivers (east of the Yamal Peninsula). This area exhibits evident warming
despite partial ice coverage of the Gulf of Ob in June and likely is driven by warmer riverine water inputs.
$CH_4$ concentrations show a clear latitudinal trend that increases towards the north. This latitudinal gradient is
weak in June and strong in September. Strong localized variations also occur in different Barents Sea regions. $CH_4$
concentrations along the Murman Current and in the (ice-covered) Kara Sea largely are below the latitudinal mean
in June, whereas west of Svalbard and in the north-central Barents Sea they are above average.
In June, $CH_4$ is depressed strongly around Svalbard and around Franz Josef Land and Novaya Zemlya. For
Svalbard, this corresponds to the SCC that hugs the shore. By September, $CH_4$ concentrations are notably different
with significantly higher $CH_4$ and a distinctly different spatial distribution. Most notable is the shift from depressed
to strongly enhanced $CH_4$ in the region to the west of Novaya Zemlya and around the Franz Josef Land archipelago.
Strong $CH_4$ enhancement also occurs in the outflow plumes of the Ob and Yenisei Rivers in the Kara Sea, around
the Taymyr Peninsula. Around Svalbard, $CH_4$ has risen to near latitudinal mean levels in September, except for
offshore north Spitsbergen and Nordaustlandet, where sea ice remains.





**3.3. Barents and Kara Seas Trends**

Across the Barents Sea, a number of different focus areas with distinct *SST* and $CH_4$ trends are identified (**Fig. 7**). These manifest significant spatial heterogeneity at the pixel scale and at the focus-area size scale. We analyze trends in aggregated-pixel "focus areas" located in key regions where *SST* change is the strongest (**Sec. 3.3**; **Supp. Fig. S6** for July and August trends).

June *SST* warming trends are fairly different from September *SST* trends (**Fig. 11**). In June, warming occurs much faster in the eastern Barents Sea, specifically, in waters affected by the Murman Coastal Current (MCC). This suggests the magnitude of atmospheric cooling during transit from the Atlantic is decreasing. Warming occurs primarily in shallow (generally less than 100-m deep) (**Fig. 11b**) waters that are generally well mixed. Sea ice is gone in this region by March-May, later in more northerly areas (**Fig. 4b**). Whereas there is no clear warming trend in July and August; a strong warming trend appears in the Kara Sea by September (**Supp. Fig. S5**). This warming trend occurs several months after the ice retreat – Kara Sea is ice-free in July (**Supp. Fig. S5**). This suggests increasing MCC penetration into the Kara Sea. Loeng (1991) reported that MCC penetration into the Kara Sea was uncommon in the middle of the 20$^{th}$ century.

FIGURE 11

More rapid warming occurs offshore of the western coast of Novaya Zemlya from June-September. This is where the Murman Current (MC) transports water towards the St Anna Trough (the dominant Barents Sea outflow), a region where shoaling is likely based on seabed topography (**Fig. 2b**) (Maslowski et al., 2004). The MC then flows (and submerges under ice and Arctic surface water) along the east shores of Franz Josef Land. Accelerated warming diminishes near the northern margin of the Kara Sea, where river outflow dominates the oceanography.

Enhanced warming also occurs to the south and to the west-northwest of Svalbard in September, following approximately the trend of the northerly fork of the NAC. In contrast, waters off east Svalbard, where the East Spitsbergen Current (ESC) transports cold Arctic waters southwards, do not exhibit a significant warming trend in September, although it does exhibit a warming trend in July. This suggests changes in the seasonal trends of PC's penetration into the Barents Sea, likely modulated by seasonal ice sheet retreat. There is no significant warming in either June or September in waters to the north of Franz Josef Land with ice-coverage persisting through September.

Overall Barents Sea atmospheric $CH_4$ is increasing (**Fig. 10C**), consistent with the global $CH_4$ trend (Nisbet et al., 2014). However it is notable that some Arctic regions exhibit significantly more rapidly increasing $CH_4$ than the global or Barents Sea trends. In June, $CH_4$ trends are largely similar in both ice-free and ice-covered areas. In near-coastal waters around Svalbard (except the east), in northern Norwegian fjords, and for the White Sea (Murmansk) where $CH_4$ growth trends are elevated.

$CH_4$ trends in September, when ice coverage has retreated to the northern edge of the Barents and Kara Seas (**Fig. 10b**), are very strongly enhanced in the East Barents Sea and the South Kara Sea. These areas coincided with areas of enhanced *SST* warming and show $CH_4$ trends almost three times as high as the general Arctic trend. In contrast, regions without enhanced warming, particularly waters affected by cold currents, exhibit the weakest $CH_4$ growth trend though slightly above the overall Barents Sea trend. In the Kara Strait between the Barents and Kara Seas, $CH_4$ increases very strongly.





Enhanced CH$_4$ growth is not evident in either June or September to the north of Svalbard, despite strong *SST*
increases; however, significant increases are evident here in August. This follows significant CH$_4$ enhancement in
July to the southeast of Svalbard. This July-August shift follows the NAC.
**4. Discussion**
**4.1. Methane transport from the seabed to the atmosphere**
In this study, we hypothesized that lower tropospheric CH$_4$ correlates both with changes in the overall water
column temperature and with *SST* changes, both satellite remote sensing products. For this analysis, we also
considered the locations of currents and trends in these currents, seabed bathymetry, prevailing winds, and available
Barents Sea, water-column temperature data – primarily the long-term Kola Section data, which due to the
importance of the Murman Current was directly relevant.
The proposed source of the atmospheric CH$_4$ anomaly is seabed seepage from either thermogenic sources, i.e.,
petroleum hydrocarbon reservoirs (Judd and Hovland, 2007), or degradation of submerged permafrost and hydrates
(Shakhova et al., 2017). Both permafrost and hydrate deposits can include both thermogenic and biogenic CH$_4$.
These emissions largely are as bubbles (Judd and Hovland, 2007) because the microbial filter generally removes
aqueous-enriched CH$_4$ fluid emissions from sediments to the water column (Reeburgh, 2003). As a bubble rises, it
loses CH$_4$ to the water column by dissolution, transporting the remainder. Larger bubbles vertically transport a
greater fraction of their contents more efficiently than smaller bubbles (Leifer and Patro, 2002). Some portion of the
dissolved fraction is transported vertically by the bubble-driven upwelling flow (Leifer et al., 2009). The remaining
fraction may either diffuse to the atmosphere by turbulence or be oxidized microbially. In the deep sea, all the
dissolved CH$_4$ likely is oxidized, where given the relevant depth scale is the winter wave mixed layer (WWML)
based on microbial oxidation timescales (Rehder et al., 1999). The Arctic WWML can extend to 100-200 m. In
shallow water (e.g., less than 20 m), most seep bubble CH$_4$ reaches the sea surface, with the portion decreasing for
smaller bubbles or depth (Leifer and Patro, 2002). For example, Leifer et al. (2017) showed that for the Laptev Sea
that ~25% of seabed CH$_4$ from 70 m reaches the sea surface directly, consistent with sonar observations of bubble
plumes reaching the sea surface (Leifer et al., 2017). CH$_4$ that is deposited deeper in the wave mixed layer (WML)
likely diffuses to the atmosphere rapidly, although stratification powerfully suppresses this transport. Storms
breakdown this stratification (Leifer et al., 2015) sparging all the dissolved CH$_4$ to the atmosphere (Shakhova et al.,
2013). Thus, CH$_4$ emissions that are deposited (by dissolution) into the WWML but below stratification may escape
many months later and distant from their seabed origin. In the process, the dissolved CH$_4$ drifts with currents, which
if driven upslope, transports the CH$_4$ to shallower depths, potentially into the WML where it can escape to the
atmosphere, termed shoaling.
In practical terms, bubble transport means that seepage extends the depth of the WWML for CH$_4$ by 50-100 m,
i.e., 150-300 m, extending the depth that storms can sparge dissolve CH$_4$ to the atmosphere. This implies, CH$_4$ from
seabed seepage over a significant fraction of the Barents Sea (**Fig. 2b**) can reach the WWML, or can be transported
by currents into shallower waters (shoaling) into the WWML.

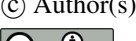



601 The above discussion was for non-oily seepage. However, where seepage arises from a petroleum hydrocarbon

602 reservoir, bubbles likely are oily. Oil slows bubble rise (Leifer, 2010) but reduces dissolution dramatically, allowing

603 their survival to far higher in the water column than non-oily bubbles (Leifer and MacDonald, 2003). Oily bubbles

604 can reach the sea surface from the deep sea due also to processes associated with hydrates, for example MacDonald

605 et al. (2010) tracked seabed seep bubbles by ROV from 1 km to the WML (upper 20 m), causing a significant

606 positive $CH_4$ anomaly in the surface waters. Given the presence of extensive proven and proposed petroleum

607 reservoirs across the Barents and Kara Seas (Rekacewicz, 2005), it is likely that some Barents Sea seepage is oily

608 bubbles, enhancing $CH_4$ transport to the sea surface.

609 Positive, localized, atmospheric $CH_4$ anomalies can reflect local seabed production and vertical transport to the

610 atmosphere, or lateral near-seabed transport and shoaling that elevates the dissolved $CH_4$ into the WML and then air

611 sea exchange into the atmosphere, or transport from a distant source. Detangling these processes leverages the

612 strength of the continuous and synoptic view of satellite datasets compared to the high spatial and temporal

613 resolution snapshot provided by field campaigns.

614 One unlikely source of $CH_4$ anomalies for the Barents and Kara Seas is atmospheric transport as there is neither

615 significant local industry, nor extensive wetlands/terrestrial permafrost nearby and downwind for the prevailing

616 winds. Note – synoptic systems can transport $CH_4$ from northern Europe or Russia to the Barents Sea area, but

617 synoptic system winds are not dominant (prevailing) and thus play a small role in a time-averaged dataset.

618 Moreover, these terrestrial sources are distant, implying extensive anomaly size scale in the Barents Sea.

619 Additionally, they would decrease with distance from northern Europe. Instead, the anomalies are localized and

620 decrease towards Europe. Although the oil production and pipeline infrastructure on the Yamal Peninsula and near

621 Kolguyev Island (**Supp. Fig. S7**) likely are strong $CH_4$ sources, they are downwind of the prevailing winds for the

622 Barents and Kara Seas.

### 623 4.2. Local versus Transported and the Importance of Shoaling

624 In this study we explore the likelihood that atmospheric $CH_4$ anomalies result from local generation and vertical

625 transport to the sea surface, versus distant lateral transport by currents prior to vertical transport to the sea surface.

626 Areas of accelerated $CH_4$ increasing trends were closely related to the path of the Murman Current as it flows

627 towards the Kara Strait rather than depth (**Fig. 11**). Both the rising seabed bathymetry and the presence of both

628 southwards and northwards currents through the Kara Strait imply strong mixing and thus transport to the

629 atmosphere. Along this path are significant offshore petroleum hydrocarbon reservoirs, which could be seeping $CH_4$

630 into the waters of the Murman Current. Further evidence for transport and shoaling is the spatial distribution of $CH_4$

631 around Kolguyev Island (north of the White Sea), which increases faster on its western side than its eastern side,

632 even though the sea to the island's east is shallower. In fact the $CH_4$ spatial pattern correlates better with shadowing

633 in the island's lee from shoaling currents, rather than with seabed depth. Prevailing winds are from the south-

634 southeast / north-northeast (Kubryakov et al., 2016), thus atmospheric transport cannot explain the pattern, which

635 would be consistent with easterly winds.





In the Kara Sea, the correlation of enhanced $CH_4$ with depth is poor, which is shallower to the north. Instead,
the location of enhanced September $CH_4$ closely matches the location of oil and gas reserves, e.g., **Supp. Fig. S7**
and Rekacewicz (2005) and also along the path of warm water transport by the Murman Coastal Current as it enters
the Kara Sea and then follows the coastline of the Kanin Peninsula. Although there are extensive oil and gas
production on the Yamal Peninsula, prevailing winds blow away from the Barents Sea. Note, the trend shows an
acceleration of $CH_4$ growth, implying increasing emissions, i.e., not steady-state seabed warming but accelerating
seabed warming. This increasing $CH_4$ growth is for September, not June, corresponding to when the water column is
warmest in the South Barents Sea (Stiansen et al., 2009). Also, the Barents Sea outflow through St Anna's Trough is
greater (about double) than June (Gammelsrød et al., 2009) when the growth in the $CH_4$ anomaly occurs (**Fig. 11d**).
The importance of this transport also is apparent in the *SST* trend with the greatest warming occurring in June in the
southeast Barents Sea (offshore the Kanin Peninsula) near the Kara Strait. This region lies to the west of the areas of
accelerating $CH_4$ growth in September near the Kara Strait. In contrast, significant *SST* warming is not observed in
September in this easternmost region of the Barents Sea.
Two other areas of accelerating $CH_4$ growth lie in the north-central Barents Sea, north of Central Bank, and
offshore northern Novaya Zemlya. These regions lie along the pathway of the Murman Current, which is an
extension of the warm NAC and the largest northwards transport of Barents Sea water (Lien et al., 2013). Water
flowing in this direction also is forced upwards – from 300-400 m to just 100 m as it crosses a sill into the St. Anna
Trough with rising seabed towards the east and towards Novaya Zemlya with water depths of tens of meters (**Fig.
2B**). Additionally, this region of accelerating $CH_4$ growth corresponds spatially to potential (unproven) gas and oil
reserves that extend across the St Anna Trough to Franz Josef Land (**Supp. Fig. S7**; (Rekacewicz, 2005)).
There are proven oil and gas fields to the south, along the path of the Murman Current, but south of the area of
accelerating $CH_4$. Again, there is good spatial correlation with these proposed reservoirs. Notably, the enhanced $CH_4$
around Franz Josef Land does not correlate with the potential reserves, but does correlate with depth and the flow of
the Murman Current, consistent with shoaling. Although some of the accelerating $CH_4$ near Novaya Zemlya could
arise from increasing local seabed emissions, seabed temperatures were below zero until 2009 (Boitsov et al., 2012),
which would imply submerged permafrost and/or hydrate deposits here have not yet degraded significantly,
supporting both a transport and a shoaling mechanism.
**4.3. Sea Surface Temperature**
The analysis shows that $CH_4$ growth from portions of the Barents and Kara Seas is accelerating faster than the
Barents Sea mean and the latitudinal mean. To some level these correlate with accelerating *SST* warming, but the
correlation is poor. One factor underlying this is the delay between *SST* warming and ocean column warming of
several months (Stiansen et al., 2009). There also appears to be a several year response time; the ~6-8 year
oscillation in the *SST* trend in the Southern Barents Sea (areas A8, A9, and A10) has a very similar timescale to the
seabed trends reported by Boitsov et al. (2012) but precedes it by ~2-4 years.
More rapid warming occurs offshore Novaya Zemlya moving northwards from June-September, where the
Murman Current transports water and the seabed topography is likely to cause shoaling. This suggests warmer





terrestrial weather is not driving Kara Sea changes as this would occur uniformly both in the south Kara Sea, which
is influenced by the Barents Sea, and the northern Kara Sea, which is influenced by river outflow. Additionally, if
increased riverine heat input was driving the trend, the greatest acceleration would be in the northern Kara Sea,
which also is shallower.

There are a number of possible explanations for why $SST$ is warming fastest in regions along the Murman

Current and NAC. One is sea-ice retreat; however, the warming occurs several months after the retreat of the sea ice.
Another is that currents are transporting warmer water; however, then warming would occur all along the pathway
of the current. Third is that stratification is becoming shallower, allowing more cooling to the atmosphere. This
would imply a weakening of storms and winds – which firstly is inconsistent with warmer $SST$, and secondly, there
is no indication that Barents Sea storminess is changing or progressing further northwards (Koyama et al., 2017).
Another possibility is that currents are strengthening. Stronger currents could relate to larger oceanographic trends.
Seabed September temperatures (Boitsov et al., 2012) do not suggest increased warmer seabed temperatures north of
Norway and Russia, but do suggest warmer seabed temperatures to the east and also along Novaya Zemlya –
suggesting a greater importance of the MC. This is consistent with the model of McClimans et al. (2000) that
currents are pushing the marginal ice zone. The warming trend suggests a strengthening of the seasonal trend in the
Barents Sea outflow, which is greater in September than June (Gammelsrød et al., 2009).

The strongest warming trend is for the shallow water off northwest Svalbard (area A4) (**Fig. 11b**), which also

exhibited the strongest acceleration of $CH_4$ growth for around Svalbard. In this area, seabed topography is nearly
level over an extensive shelf with depths in the range 250-400 m. Where the shelf falls off sharply, rising sea
temperatures will minimally induce hydrate destabilization. In contrast, where the shelf falls off very gently, small
temperature increases shift extensive areas of seabed from below to above the hydrate stability field. This area is
immediately to the north of the area where several researchers have identified extensive seabed seep $CH_4$ emissions
(Mau et al., 2017; Myhre et al., 2016; Westbrook et al., 2009). The most likely explanation is a strengthening of the
West Spitsbergen Current, discussed below, and shifts in the Barents Sea Front.
**4.4. Implications for Svalbard Area Emissions**

There are few atmospheric and ocean $CH_4$ data for the Barents Sea and surrounding areas, the most prominent

being associated with $CH_4$ seepage off Spitsbergen, Svalbard, located immediately south of focus area A4. Studies
to date have been in early summer; Mau et al. (2017); Myhre et al. (2016) who made measurements in the
atmosphere and water column; Westbrook et al. (2009) reported sonar observations of seep bubbles for August-
September, and slightly elevated aqueous $CH_4$ in surface waters immediately above the bubble plumes. All
concluded transport to the atmosphere was not significant, attributed to trapping of dissolved $CH_4$ below a sharp
pycnocline. It is important to note that with respect to the overall Barents Sea area $CH_4$ anomaly, the Svalbard area
is far less important than around Franz Josef Land, off the west coast of Novaya Zemlya, and the north-central
Barents Sea (**Fig. 10**).

Both $SST$ and $CH_4$ in June (**Fig. 10**) and July (**Supp. Fig. S5**) for west Svalbard show that much of the area of

active seepage was inshore of the Barents Sea Front, and thus under the cooling Arctic waters of the Spitsbergen



Coastal Current (SCC), supported by reported salinity data (Mau et al., 2017). Although the *SST* remains suppressed
off Spitsbergen in September, and extends further offshore, $CH_4$ concentrations no longer are depressed compared to
Atlantic water further offshore, i.e., greater transport to the atmosphere. Such transport would not be expected
downcurrent (north) of the bubble plumes observed by the early fall cruise reported in Westbrook et al. (2009).
Although data indicate these seeps do not contribute to summer atmospheric $CH_4$, this study suggests that the
emitted $CH_4$ likely is reaching the atmosphere far downstream where currents shoal. Interestingly, Mau et al. (2017;
Fig. 3) show data that could be interpreted as shoaling with elevated aqueous $CH_4$ forced shallower by the north-
flowing SCC, rising as it crosses onshore-offshore aligned subterranean ridges. Focus area A4 shows strong increase
in $CH_4$ from 2005-2015 (the strongest of the focus areas (**Table 1**) and in increasing *SST* over this time period,
consistent with shoaling. Larger acceleration of $CH_4$ growth is observed north of Spitsbergen in June (**Fig. 11c**),
which is the most likely location for shoaling based on detailed Svalbard bathymetry and currents (**Supp. Fig. S2**).
Specifically, this is where some of the warm West Spitsbergen Current mixes with the cold, Spitsbergen Coastal
Current (SCC) that would be $CH_4$ enriched from seabed seepage, and then flows over relatively shallow seabed
towards the Hinlopen Strait. To summarize, although there is evidence of increasing downcurrent $CH_4$ transport to
the atmosphere downcurrent of seepage off West Svalbard after shoaling, it is not significant with respect to overall
Barents Sea emissions.
There is evidence of acceleration in the $CH_4$ growth nearshore off western Svalbard in June, but not in
September (**Fig. 11**) when $CH_4$ growth acceleration lies in the further offshore waters that are impacted by the warm
WSC. Trends in *SST* also suggest a weakening of the Percey Current in June and more so in September. Given that
from June to September the SCC extends further offshore, this suggests WSC control. Similarly, the WSC eastwards
leg that crosses Nordaustlandet is driving a rapid increase in *SST* in September and likely relates to the increased
$CH_4$ trend.
**4.5. Ice-Free Barents Sea**
The ice coverage trend shows that northeast Barents and southern Barents Sea already are ice-free or near ice-
free year round, whereas northwest Barents Sea (around Franz Josef Land and St. Anna Trough) remains ice-
covered for about half the year. The ice coverage trends (**Fig. 6**) suggest most of the Barents Sea will be ice free,
year-round in another decade and a half, circa 2030. This is comparable to the 2023-2036 estimate of Onarheim and
Årthun (2017; Fig. 3), which also notes that the current decreasing trend lies outside the oscillation envelope since

1850.

This has implications for the Barents and Kara Seas ecosystems, and follows changes that have been
documented across the Arctic in satellite remote sensing of phytoplankton concentration (Arrigo et al., 2008; Arrigo
and van Dijken, 2011; Kahru et al., 2011) and *in situ* studies (Grebmeier et al., 2015; Grebmeier et al., 2006). One
example is a significant northwards shift (5° over 20 yrs.) of phytoplankton blooms (Neukermans et al., 2018). Ice
cover changes play a key role. For example, primary productivity increases in the northern Barents and Kara Seas
(**Fig. 2**) are considered caused by decreased ice cover (Slagstad et al., 2015), which has driven changes in the higher
trophic levels of the pelagic and benthic community (Grebmeier et al., 2015).





The Barents Sea is a marginal sea between the temperate Norwegian Sea and the Arctic Ocean and thus is the
conduit through which lower-latitude oceanic heat is transmitted to the Arctic Ocean (Onarheim and Årthun, 2017).
Given the significant role the Barents Sea plays in overall Arctic ice loss - fully 25% of the loss is attributed to the
Barents Sea, which comprises 4% of the Arctic Ocean and marginal seas (Smedsrud et al., 2013), implications will
be significant for weather including at lower latitudes, and the marine ecosystem. Seemingly counter-intuitively, the
reduction of sea ice increases the upwards surface heat flux as ice has an insulating effect. Thus ice-loss somewhat
stabilizes Arctic Ocean ice, particularly during winter (Onarheim and Årthun, 2017) and may even lead to growth of
ice in the Arctic and Northern Greenland Seas. Still, the data herein showed a progressive weakening of the Percey
Current, which will continue to cause ice loss off east Svalbard and warming of these waters. This agrees with
Alexander et al. (2004) who concluded that the (semi-stationary due to bathymetry) Polar Front has shifted to
domination of Atlantic water over Arctic waters.
As already seen, though, the progression of ice loss in the south and east Barents Sea along the pathway of the
Murman Coastal Current has led to a progressive loss of ice in the south Kara Sea. Thus, the balance between the
two processes – heat loss to the atmosphere from and progressive transport of heat by currents to the Kara Sea are
clearly shifting towards warmer. The implications of decreasing ice coverage in the shallow Kara Sea are significant
with respect to $CH_4$ emissions – the area is rich in hydrocarbon resources that currently are sequestered (albeit data
show already poorly) under submerged permafrost that will continue to degrade, while warming seabed
temperatures will enhance microbial degradation of the vast organic material deposited over the millennia by the Ob
and Yenisei Rivers. Thus, the already significant importance of Arctic $CH_4$ anomaly from the Kara Sea will
accelerate due to feedbacks from an ice-free Barents Sea.
**4.6. Future research**
Long-term oscillations with a timescale of 6-8 years were identified (e.g., **Fig. 9**); however, the length of the
dataset (13 years) is too short to investigate this in detail. Extending the analysis to include more recent data (say
through 2020) would span a full 2 1/2 cycles and allow investigation of correlations with other driving
oceanographic atmospheric cycles, such as the NAO. This would be particularly valuable given that recent data
show that the most recent two years are the most extreme in terms of Barents Sea ice coverage Oziel et al. (2016)
and $CH_4$ anomaly (**Supp. Fig. S8; Supp. Video**). Extending the analysis forward in time clearly would provide
greater insights into the complex relationship between currents and $CH_4$ emissions. In this regard, the long $CH_4$ time
series planned to be collected by the IASI satellite series through the 2030s (Onarheim and Årthun, 2017) will be
invaluable.
Additionally, there is clearly need for these data to be incorporated into coupled atmospheric-oceanographic-ice
models to understand in greater detail the processes underlying the changes, improving the ability to forecast trends
in Arctic marine greenhouse gas emissions. Currently the strong and growing $CH_4$ anomaly from Novaya Zemlya
and Franz Josef Land are the strongest in the Arctic, yet are not yet incorporated (or identified) in inversion models,
e.g., Crevoisier et al. (2014). This identifies a key strength of satellite data, which can identify sources that are not
part of an apriori.



Finally, as part of this study, changes in chlorophyll were investigated with respect to changes in *SST* and $CH_4$. These relationships need to be evaluated in future research to tie the dramatic changes in the Barents and Kara Sea ecosystem to physical changes in oceanography, leveraging the strengths of satellite data.

**5. Conclusion**

In this study, the synoptic, repeat nature of satellite data was used to investigate the relationship between currents, and trends in sea surface temperature, ice extent, and methane ($CH_4$) anomaly for the Barents and Kara Seas for 2003-2015. Large positive $CH_4$ anomalies were discovered around Franz Josef Land archipelago and offshore west Novaya Zemlya in September, in areas where currents shoal, with far smaller $CH_4$ enhancement around Svalbard, again, strongest where currents likely shoal, downcurrent of seabed seepage. This highlights a major strength of satellite data: Identification of sources that are not part of an apriori in inversion models.

The strongest *SST* increase was southeast Barents Sea in June due to strengthening of the warm Murman Currents (an extension of the Norwegian Atlantic Current) and in the south Kara Sea in September, whereas the cold Percey Current weakened. These two regions also exhibit the strongest $CH_4$ growth acceleration as well as around Franz Josef Land. Likely sources are $CH_4$ seepage from extensive oil and gas reservoirs underlying the central and east Barents Sea and Kara Sea; however, the spatial pattern was poorly correlated with depth and best correlated with strengthening current that shoal.

If current trends continue, then heat flows to the Barents Sea and Kara Sea by strengthening currents will lead to an ice free Barents Sea free in about 15 years, while driving seabed warming and enhanced $CH_4$ emissions, particularly from areas where currents drive shoaling.

**Acknowledgements:** The research was supported by a grant from NASA ROSES2013: "A.28, THE SCIENCE OF TERRA AND AQUA: Long-term Satellite Data Fusion Observations of Arctic Ice Cover and Methane as a Climate Change Feedback." We thanks Vladimir Ivanov, Arctic and Antarctic Research Institute, for organizing the NABOS cruise and Cathrine Lund Myrhe, Norwegian Air Research Institute (NIILU), for calibration gas used during the NABOS cruise.



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



**TABLES**

**Table 1**. Slopes of *SST* (°C yr$^{-1}$), CH$_4$ (ppb yr$^{-1}$), and CH$_4$' (ppb yr$^{-1}$) for focus boxes. [a]

| Box | *SST* | CH$_4$ | CH$_4$ | CH$_4$' (Barents)[b] | CH$_4$' (Arctic)[c] |
|---|---|---|---|---|---|
| | 2003-2015 | | 2003-2015 | 2005-2015 | 2003-2015 | 2003-2015 |
| A1 | 0.102 | 3.35 | 3.26 | 0.179 | 0.0750 |
| A2 | 0.0319 | 3.49 | 3.38 | 0.267 | 0.213 |
| A3 | 0.00178 | 3.19 | 3.17 | -0.0185 | 0.00574 |
| A4 | 0.0867 | 3.37 | 3.60 | 0.310 | 0.391 |
| A5 | 0.0279 | 3.10 | 3.22 | 0.0105 | 0.0319 |
| A6 | 0.00259 | 3.07 | 3.24 | -0.0123 | 0.0548 |
| A7 | 0.0323 | 3.06 | 3.27 | -0.0460 | -0.119 |
| A8 | 0.0552 | 3.11 | 3.35 | 0.0642 | -0.0544 |
| A9 | 0.145 | 3.20 | 3.44 | 0.103 | 0.109 |
| A10 | 0.0527 | 3.32 | 3.51 | 0.122 | 0.0613 |

[a] *SST* – Sea Surface Temperature, CH$_4$' – methane anomaly.

[b] CH$_4$' relative to the Barents Sea

[c] CH$_4$' relative to the Arctic Ocean





**FIGURE CAPTIONS**

**Figure 1 a)** Arctic and sub-arctic methane ($CH_4$), 0.5° gridded, 0-4 km altitude, 2016, from the Infrared
Atmospheric Sounding Interferometer (IASI); mountainous regions blanked. Data were filtered as in
Yurganov and Leifer (2016a). Data key on panel.
**Figure 2 a)** Map of the Arctic Ocean, showing study area (Blue Square) and average January and
September 2003-2015 ice extent. **b)** Bathymetry of the study area (87.468 N, 1.219E; 72.056N, 0.173E;
63.008N, 48.05E; 69.707N, 82.793E) from Jakobsson et al. (2012). Dashed line shows approximate
Barents Sea boundaries. Star shows scoping study pixels location. Depth data key on panel.
**Figure 3**. Comparison of the sea surface temperature (*SST)* and methane ($CH_4$) for 2003-2015 for pixels
between Franz Josef Land and Novaya Zemlya (**Fig. 2b, Star, Supp. Table 1, Box A2**). Red diamonds
show *SST* and $CH_4$ averages within the study area. Blue and green ovals highlight pixels with different
$CH_4$ trends for *SST* (all $CH_4$), and ($CH_4$>1925 ppb), respectively.
**Figure 4. a)** Currents for Barents and nearby seas, bathymetry features, and focus-area locations. Green,
red, and blue arrows are coastal, warm Atlantic origin, and cold polar currents, respectively. Broken lines
illustrate current subduction. Bathymetry from Jakobsson et al. (2012). **b)** Monthly ice extent for 2015.
Focus study boxes (numbered); coordinates listed in **Supp. Table S1**.
**Figure 5. a)** Surface in situ methane ($CH_4$) during northward Barents Sea transect on the *R/V Akademik*
*Federov* for 21 Aug. 2013. Also shown is the 300-m depth contour and edges of the Murman Coastal
Current, from pinru(http://www.pinro.ru/labs/hid/kolsec1_e.htm). Data key on figure. **b)** $CH_4$ profiles
during northerly and southerly transits, labeled.
**Figure 6.** Ice-free months from 2003 to 2015 for focus boxes for **a)** Northern Barents (A1-A3), **b)**
Northwest of Barents (A4-A6), and **c)** Southern Barents (A7-A10). Box names on panels. See **Fig. 3c** and
**Supp. Table S1** for locations.
**Figure 7.** Sea surface temperature (*SST*) time series for 2003 to 2015 for focus box areas **a)** Northern
Barents (A1-A3), **b)** Northwest of Barents (A4-A6), and **c)** Southern Barents (A7-A10). Annual values
are average of all months, generally May-October, which are ice-free. Box names on panel a. Data key on
figure.
**Figure 8.** Focus study area methane ($CH_4$) trends, 2003 to 2015 for **a)** Arctic Ocean study boxes, **b)**
Northwest of Barents study boxes, and **c)** Barents Sea focus study boxes. Annual data and 3 year, rolling-
average data shown. Anomaly is relative to entire Barents Sea. Data key on figure.
**Figure 9.** Trends for focus area A8 (north of Murman) for 2003 to 2015 for $CH_4$' and *SST*. Data key on
figure.
**Figure 10.** Mean values for 2003 to 2015 of sea surface temperature (*SST*) for **a**) June and **b**) September.
Mean methane ($CH_4$) concentration for **c**) June and **d**) September. Median ice edge for same period is
shown. Years with reduced ice extent contribute to values of *SST* north of this ice edge. Data key on
figure.
**Figure 11.** Linear trends for 2003 to 2015 of sea surface temperature (*dSST*/*dt*) for **a**) June and **b**)
September. Methane concentration trend (*d*$CH_4$/*dt*) for **c**) June and **d**) September. ND – not detectable –
failed statistical test. Blue, black dashed lines shows 100 and 50 m contour, respectively. Data key on
figure.





FIGURES

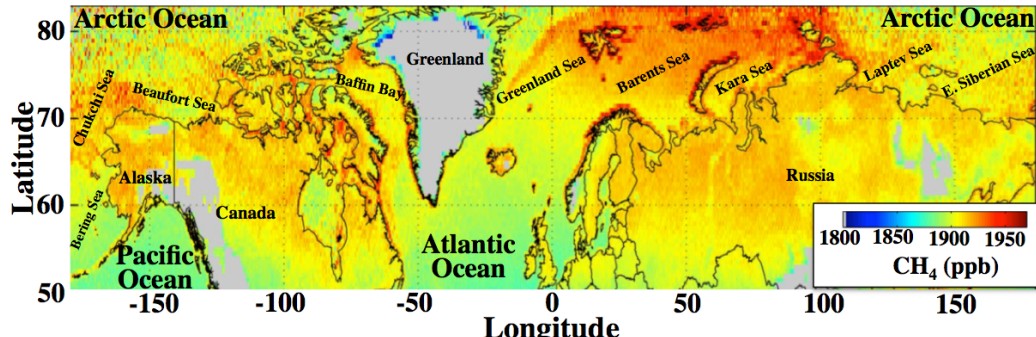


**Figure 1** Arctic and sub-arctic methane ($CH_4$), 0.5° gridded, 0-4 km altitude, 2016, from the Infrared
Atmospheric Sounding Interferometer (IASI); mountainous regions blanked. Data were filtered as in
Yurganov and Leifer (2016a). Data key on panel.




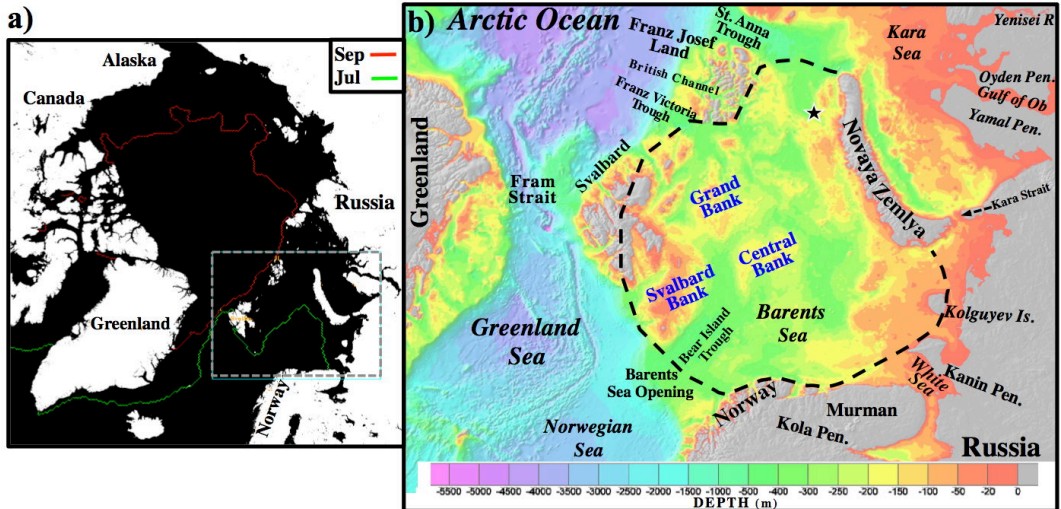

**Figure 2 a)** Map of the Arctic Ocean, showing study area (Blue Square) and average January and September 2003-2015 ice extent. **b)** Bathymetry of the study area (87.468 N, 1.219E; 72.056N, 0.173E; 63.008N, 48.05E; 69.707N, 82.793E) from Jakobsson et al. (2012). Dashed line shows approximate Barents Sea boundaries. Star shows scoping study pixels location. Depth data key on panel.



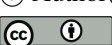

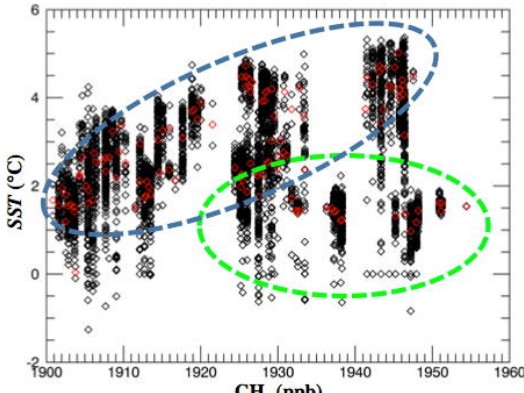

**Figure 3**. Comparison of the sea surface temperature (*SST*) and methane ($CH_4$) for 2003-2015 for pixels between Franz Josef Land and Novaya Zemlya (**Fig. 2b, Star, Supp. Table 1, Box A2**). Red diamonds show *SST* and $CH_4$ averages within the study area. Blue and green ovals highlight pixels with different $CH_4$ trends for *SST* (all $CH_4$), and ($CH_4$>1925 ppb), respectively.



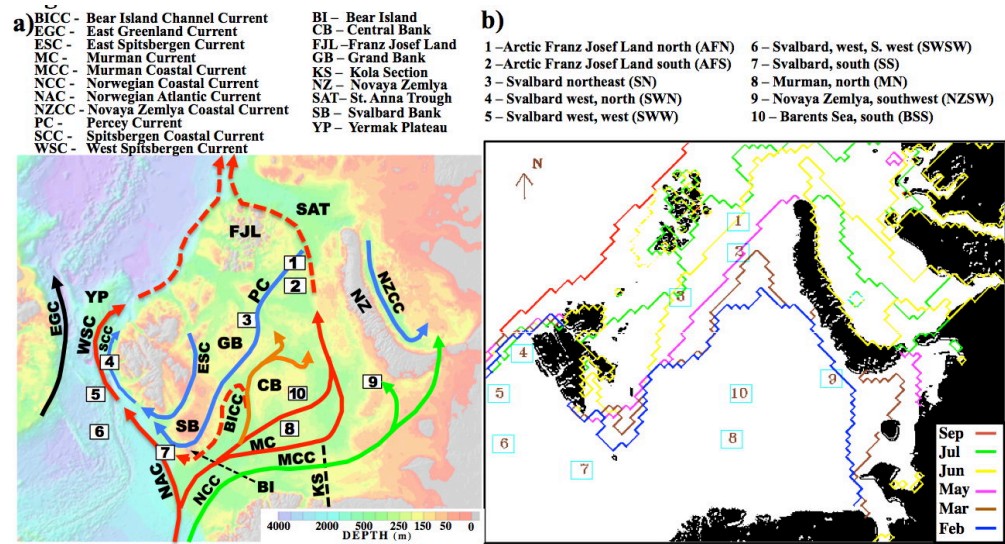


**Figure 4. a)** Currents for Barents and nearby seas, bathymetry features, and focus-area locations. Green,
red, and blue arrows are coastal, warm Atlantic origin, and cold polar currents, respectively. Broken lines
illustrate current subduction. Bathymetry from Jakobsson et al. (2012). **b)** Monthly ice extent for 2015.
Focus study boxes (numbered); coordinates listed in **Supp. Table S1**.







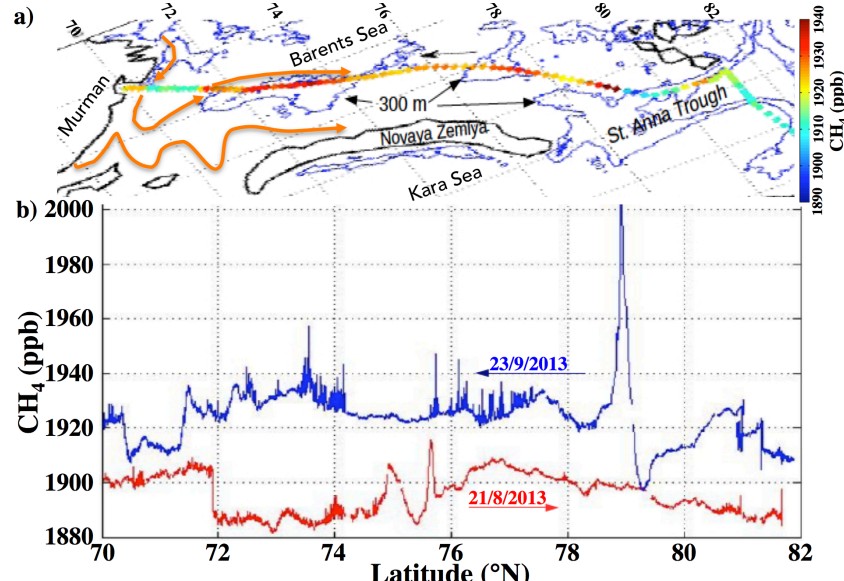

**Figure 5. a)** Surface in situ methane ($CH_4$) during northward Barents Sea transect on the *R/V Akademik Fyodorov* for 21 Aug. 2013. Also shown is the 300-m depth contour and edges of the Murman Coastal Current, from PINRU (http://www.pinro.ru/labs/hid/kolsec1_e.htm). Data key on figure. **b)** $CH_4$ profiles during northerly and southerly transits, labeled.



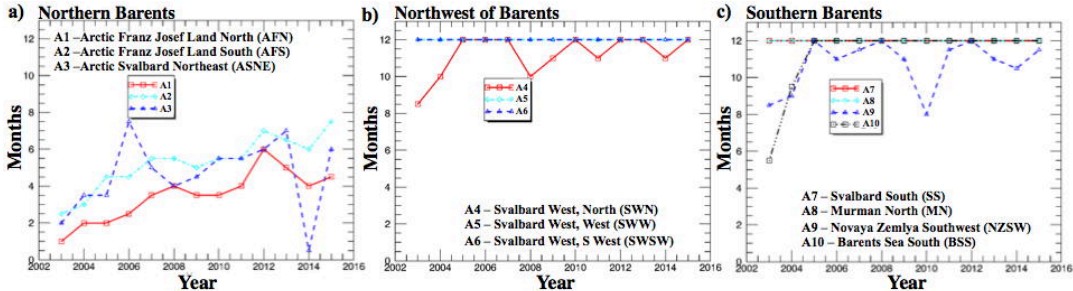

**Figure 6.** Ice-free months from 2003 to 2015 for focus boxes for **a)** Northern Barents (A1-A3), **b)** Northwest of Barents (A4-A6), and **c)** Southern Barents (A7-A10). Box names on panels. See **Fig. 3c** and **Supp. Table S1** for locations.





1334

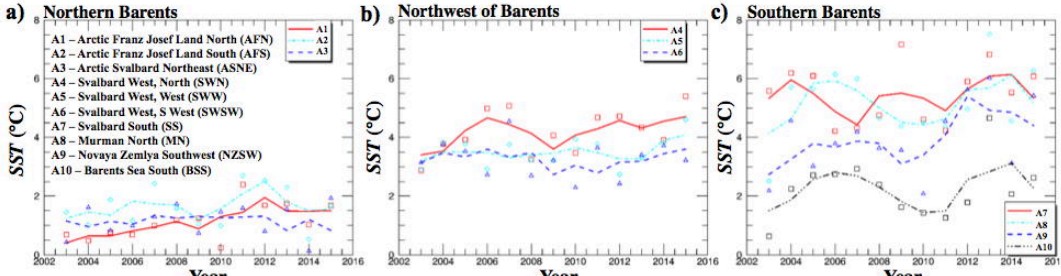

1335

**Figure 7.** Sea surface temperature (*SST*) time series for 2003 to 2015 for focus box areas **a)** Northern
Barents (A1-A3), **b)** Northwest of Barents (A4-A6), and **c)** Southern Barents (A7-A10). Annual values
are average of all months, generally May-October, which are ice-free. Box names on panel a. Data key on
figure.





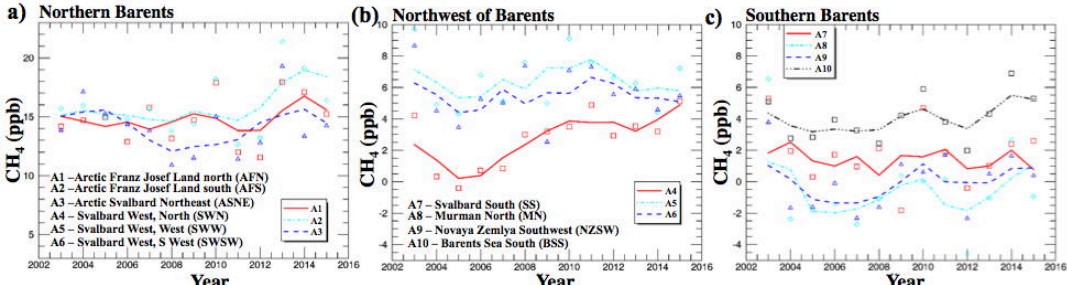

**Figure 8.** Focus study area methane ($CH_4$) trends, 2003 to 2015 for **a)** Arctic Ocean study boxes, **b)** Northwest of Barents study boxes, and **c)** Barents Sea focus study boxes. Annual data and 3 year, rolling-average data shown. Anomaly is relative to entire Barents Sea. Data key on figure.



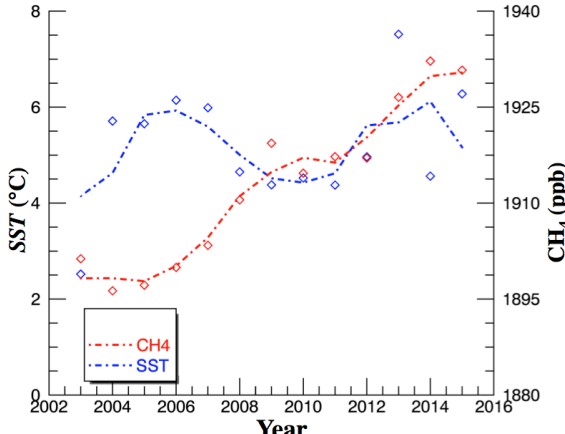


**Figure 9.** Trends for focus area A8 (north of Murman) for 2003 to 2015 for CH$_4$' and *SST*. Data key on
figure.




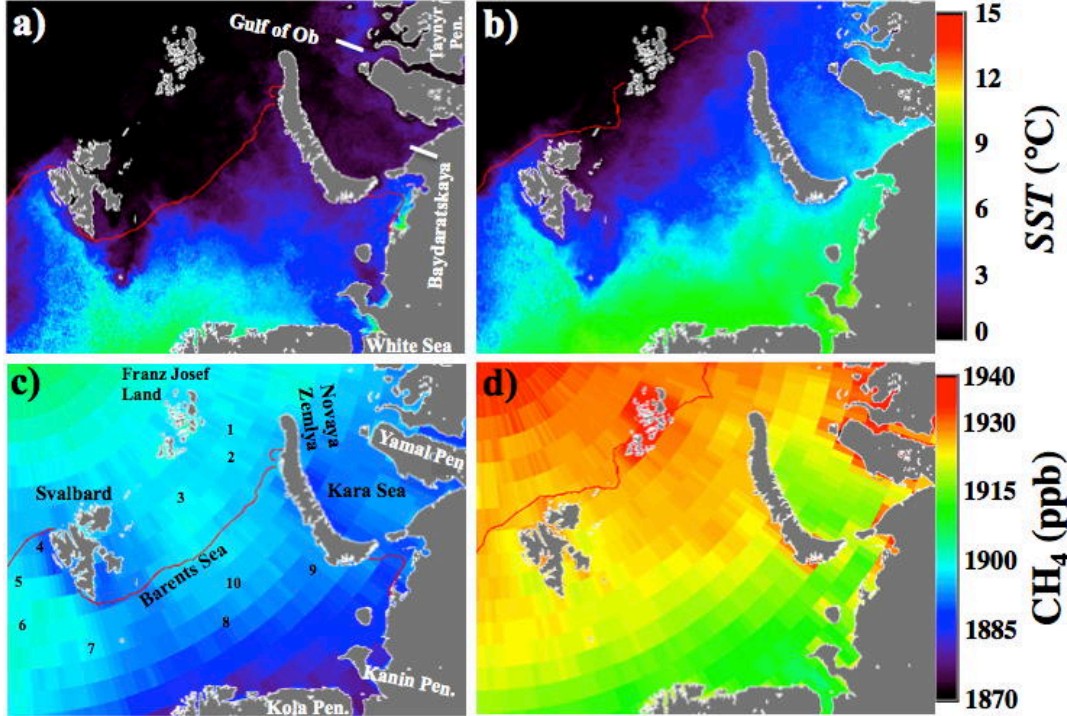

**Figure 10.** Mean values for 2003 to 2015 of sea surface temperature (*SST*) for **a**) June and **b**) September.
Mean methane (CH$_4$) concentration for **c**) June and **d**) September. Median ice edge for same period is
shown. Years with reduced ice extent contribute to values of *SST* north of this ice edge. Data key on
figure.





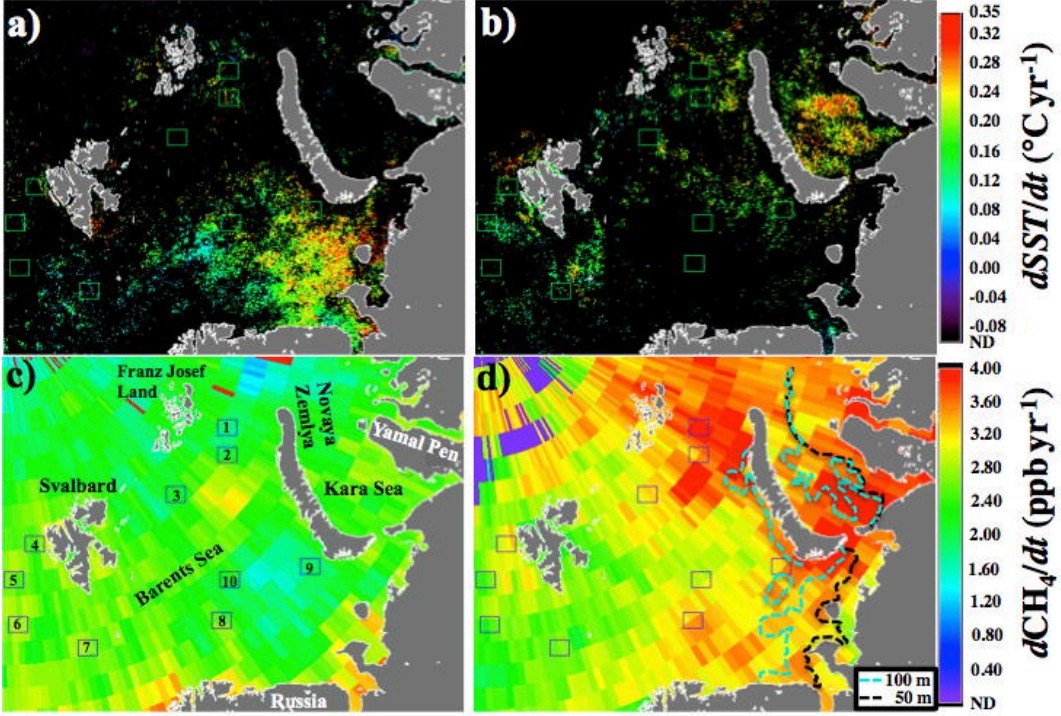

**Figure 11.** Linear trends for 2003 to 2015 of sea surface temperature ($dSST/dt$) for **a**) June and **b**) September. Methane concentration trend ($dCH_4/dt$) for **c**) June and **d**) September. ND – not detectable – failed statistical test. Blue, black dashed lines shows 100 and 50 m contour, respectively. Data key on figure.