# Peer review of "Satellite ice extent, sea surface temperature, and atmospheric methane trends in the Barents and Kara seas"

_The Cryosphere, 2018_

## Author Comment (AC1) · 27 May 2018

We would like to correct the naming of a current in Supplemental Figure S1, which was mistakenly identified as BICC - the Bear Island Channel Current, but which should instead be the Bear Island Current (BIC).

[Figure]

[Figure]

| | |
|---|---|
| BIC | Bear Island Current |
| ESC | East Spitsbergen Current |
| PC | Percey Current |
| SC | Sørkapp Current |
| SCC | Spitsbergen Coastal Current |
| WSC | West Spitsbergen Current |

**Fig. 1.** Rev Supp Fig. S1

---

## Referee Comment (RC1) · Anonymous Referee #1 · 1 Jun 2018

Review of "Review of Leifer et al: Methane trends in the Barents and Kara Seas"

The submitted paper presents new observations of methane from the Barents and Kara Seas, and also other Arctic areas. Large anomalies are documented in some areas, and these new observations appear fine, and probably deserves to be published.

In addition to the methane data the paper also aims at documenting changes in the sea ice cover, and draw conclusions about changes in ocean currents based on SST data. This effort is not well focused, and there are a number of fundamental problems listed below.

So I am sorry to say that I cannot recommend publication of this paper in any form

close to this submitted version. My advise is to extract only the Methane data, present and discuss that, but stop there.

Perhaps the most fundamental problem is already clear in the first 4 words in the abstract: "Long-term (2003-2015) ... ". Available observations for the Barents Sea go back at least 100 years, and the Kola section data is available through the PINRO institute, also cited in the paper (Page 17, line 574).

There are large oscillations in this very coupled air-ice-ocean system, om many time-scales, up to at least 50-60 years. Stating that 13 years of data is "long-term" in such a system reveals a fundamental lack of physical understanding, and also neglects the most basic findings in this area. Some essential papers are cited in this regard, but still the short time-series made available "suggests" a number of things to the authors.

There are also fundamental data missing for drawing the conclusions; one cannot deduce changes in ocean currents without observations of flow, or vertical structure of salt and temperature. Likewise - when concluding on changes in sea ice cover and SST, data on heat fluxes to that atmosphere is fundamental, and not included here.

Other Major Issues:

Length: The Results starts on page 12. There is a very long attempt at summarizing different aspects, but not a good one. There is also 4 pages of "Methods" with text that should have been in the introduction (Section 2.3 Settings). So overall this is just summarizing other peoples work, and it is also not a good summary. At least here "Long term" is used in the correct way (Page 10, line 350), stating 1905 onwards.

Figures. The figures are of a poor quality. Legends are too small (Fig. 7 – for example), There are a number of errors – the worst probably that the East Greenland Current flows northwards! In Fig. 4 a). Figure captions are not clear, and some Figures are missing (Fig. 3 c) stated in Figure 6 caption. Trends – as used in Figure 9 caption does not show trends, but a time series.

The Language is poor – see examples below.

Some citations are used wrongly.

There is an overall lack of physical understanding – or perhaps just sloppy language. One Example from Page 4, line 131: "Sea ice significantly reduces surface albedo in the Arctic summer". Sea ice INCREASES surface albedo. Along the same lines the Barents Sea has lost sea ice during winter mostly, and is certainly not the place that has the largest summer ice loss. Sea ice data is available back to 1850 (Walsh et a. 2017), and the southern Barents Sea has also been ice free during all that time.

The contribution from the atmosphere is totally ignored – but is of vital importance (Lee et al. 2017)

The conclusions drawn are not supported by the data made available.

A few of the (too many) Minor Issues:

Figure 1: Use Polar Stereographic Map. The Arctic is the focus here.

Figure 2: Legend states Jul, caption states Januay.

Figure 3: Red diamonds shows what kind of averages?

Figure 4: What is the citation for the currents? Northwards EGC!

Figure 5: PINRU in Caption. The 5 a) Map should be similar to the other maps.

Figure 6: Missing Fig. 3 c). Are the A areas the same as the numbers in Fig. 4?

Figure 7: Too small caption.

Figure 8: Legend in wrong figures. A - areas are what?

Figure 9: Trends are not single points. No trends on figure.

Line 78: The largest is loss in the WINTER – not SUMMER. (Onarheim et al 2018).

[Figure]

Line 134 – 135: Summer or Winter? A warm annual-mean surface is mostly driven by increased AW inflow, preventing sea ice formation (Årthun et al 2013).

Line 143: winter mixed layer extends to bottom? Citation needed!

Line 435: The Barents Sea is included in the Arctic Ocean.

Line 439 – 446: Lacking fundamental physics. Ice advection and surface fluxes are not even mentioned.

Line 483 – 484: . . . "likely arises from variations in the strength and location of the MC". You have shown no data to support this statement. It is just a wild guess.

Line 489 – 490: There is no importance AW heat visible in Fig. 9.

Line 515 – 516: What about solar forcing – which you state is important in the Barents Sea. Here it would probably be the major contributor – not mentioned at all.

Line 536: No data on vertical mixed water.

Line 539 – 541: There is a fundamental lack of physics discussing atmospheric forcing. This is probably the major cause of the ice loss here (Lee et al, 2017).

Line 550 – 554: Just speculations – without surface flux estimates.

Line 574: No vertical data is shown, plenty is available.

Line 614 – 622: Alos wind forcing is not included, analyzed or discussed properly.

Line 682: Sloppy language: "Stronger currents could relate to larger oceanographic trends." What trends?

Line 686: Sloppy language: "currents are pushing the marginal ice zone."

Line 731: The southern Barents Sea has been ice free since 1850 (Walsh, et al 2017).

Line 751: Percey Current weaking? No – you show an SST cooling.

[Figure]

Line 757: "Clearly shifting?" - No – you do not shown any atmospheric fluxes.

Line 772: No methane discussion in Onarheim and Årthun (2017).

Line 779: . . . part of an apriori what?

Line 790 and 796: Again mistaking as increased SST signal with a strengthening current without further evidence.

A few suggested papers too read:

Lee at al (2017) Revisiting the cause of the 1989-2009 Arctic Surface Warming. GRL. 44.

Årthun et al (2013) Quantifying the Influence of Atlantic Heat on Barents Sea Ice Variability and Retreat J Clim., 25, doi:10.1175/JCLI-D-11-00466.1, 2012

Onarheim et al (2018) Seasonal and regional manifestation of Arctic sea ice loss, J. Clim. 31, doi:10.1175/JCLI-D-17- 0427.1

Walsh et al. (2017) A database for depicting Arctic sea ice variations back to 1850. Geogr. Rev., 107, 89–107, https://doi.org/10.1111/j.1931-0846.2016.12195.x.

---

## Referee Comment (RC2) · Anonymous Referee #2 · 12 Jun 2018

Review of "Satellite ice extent, sea surface temperature, and atmospheric methane trends in the Barents and Kara Seas" by Ira Leifer et al.

General comments

The submitted paper looks rather interesting. The authors carried out study in considering one of the current warming consequences, namely: increasing methane concentrations. They made a detailed enough review of the relevant literature and presented rather detailed analysis of the subject. However, the paper needs reworking in some ways listed below.

The introduction and methods seem to be too long. These two chapters take about 11

[Figure]

pages of text compared to about 10 pages for the results and discussions.

Subchapter 2.3 Setting looks redundant, especially for the chapter describing materials and methods, except the very last paragraph.

Some inaccurate using of Barents Sea currents names take place in the paper (see below).

Drawing conclusions about variability in currents taking into account only variability of SST is incorrect.

Most figures are too small, of a poor quality and, as a result, hardly readable.

Specific comments

Line 26: The Murman Coastal Current flows mainly eastward. The Murman Current also flows eastwards towards Novaya Zemlya, and in the eastern part, the flow (usually called as the Novaya Zemlya Current) is generally northeastwards along the slopes of the North Kanin, Goose and Novaya Zemlya Banks.

Line 27: SST depends on both currents and air temperature. Air temperature plays a key role in SST variability in shallow waters of the south-eastern Barents Sea and southern Kara Sea. Heat fluxes between ocean and atmosphere are important there.

Line 55 and on: There is one confusing term in the paper, namely: albedo. Usually, surface albedo is defined as the ratio of irradiance reflected to the irradiance received by a surface, and ranges from about 0.9 for fresh snow to about 0.04 for charcoal, one of the darkest substances. In the paper, albedo seems to be defined as the ratio of irradiance absorbed to the irradiance received, doesn't it?

Line 60: The word "Arctic" is unnecessary in the phrase "the East Siberian Arctic Sea".

Line 330: Novaya Zemlya is an archipelago, not an island, and includes two large islands (Southern and Northern) and lots of small.

[Figure]

Line 327: The phrase "The relatively shallow (230-m average depth) Barents Sea (Fig. 4) is characterized by a deep Arctic shelf. . ." is ununderstandable.

Line 335 and on: The Norwegian, Greenland, Barents, Kara and some other seas are included in the Arctic Ocean. Therefore, it is better to say the Arctic Basin (the central area of the Arctic Ocean beyond all these seas) in this context.

Line 338: There is no current in the Barents Sea called "Bear Island Channel Current". There are warm North Cape current flowing eastwards in the Bear Island Trough and cold Bear Island Current flowing southwest along the southeastern slope of the Spitsbergen Bank.

Line 340: The Grand bank mentioned in the paper is commonly known as the Great bank.

Lines 385–387: There is a repetition in this place: "cooler surface water . . . flows in NZCC and exit through the Kara Strait". The NZCC abbreviation expansion needs being done at the first mention in line 385.

Line 403: The cold Bear Island Current flows there, not the Persey Current that is located further northeast.

Line 404: Both focus areas (A8 and A10) are only influenced by Atlantic waters, the Persey Current does not influence them (this is clearly seen in picture 4a).

Line 418: The Saint Anna Trough is a correct name.

Line 418: The focus areas are shown in figure 4, not 3.

Line 430: According to bathymetry shown in figure 4a, focus area 7 covers the shelf slope, and focus area 8 covers depths of more than 200 m. They do not seem to cover banks.

Lines 435 and 451: All three northern focus areas (A1–A3) are located south of the FJL, none of them are east of the FJL.

Line 450: According to figure 7, trends are almost absent in some areas (A3 and A6, for example).

Line 453: This is a moot point that trends in SST are consistent with strengthening or weakening of currents. Advection of waters with higher temperatures and local conditions effect these trends as well. For example, the volume flux into the Barents Sea was rather low from 2007 to 2015 (ICES. 2017. Report of the Working Group on the Integrated Assessments of the Barents Sea. WGIBAR 2017 Report 16-18 March 2017. Murmansk, Russia. ICES CM 2017/SSGIEA:04. 186 pp.).

Lines 483–484: Measured volume fluxes through the Barents Sea Opening (see reference above) are weakly consistent with this statement.

Lines 489–490: The mentioned importance is not clear from figure 9 at all.

Line 491: It is ununderstandable what current is mentioned (BIC). Along the eastern coastline of the Svalbard, the cold East Spitsbergen Current only flows southwards.

Lines 498–499: This is the Bear Island Current, not the Persey Current.

Line 540: The Kara Sea is ice-free in September and almost ice-free in August but not in July (even during the last warm period). In July, the northern and northeastern parts of the Kara Sea as well as the area close to the eastern coastline of the Novaya Zemlya Archipelago are still ice-covered often.

Lines 614–617: The authors need to pay more attention to winds over the Barents and Kara Seas; the role of atmospheric circulation seems to be underestimated.

Line 682: The evidence of the statement that currents are strengthening is not provided. The phrase "larger oceanographic trends" is ununderstandable.

Lines 731 and 732: There is some mess in using "northwest" and "northeast". The FJL and St. Anna Trough are in the northeastern Barents Sea, not in the northwestern. The southwestern and southern parts of the Barents Sea are ice-free year round, but

not the northeastern part.

Figure 4a: The East Greenland Current looks like having the wrong direction. This current is flowing southwards (Skjoldal H.R. (Ed.). 2004. The Norwegian Sea Ecosystem. Tapir Academic Press, Trondheim. 559 pp.; Dickson R.R., Meincke J., Rhines P. (Eds). 2008. Arctic–Subarctic Ocean Fluxes: Defining the Role of the Northern Seas in Climate. Springer, Dordrecht. 736 pp.). The detailed description of the Barents Sea currents is also presented in: Jakobsen T., Ozhigin V.K. (Eds). 2011. The Barents Sea: ecosystem, resources, management: Half a century of Russian-Norwegian co-operation. Tapir Academic Press, Trondheim. 825 pp. (scanned version of this book is available at https://brage.bibsys.no/xmlui/handle/11250/109444).

Figure 6: The caption mentions figure 3c that is absent.

Figures 7, 8 and 9: There is no explanation for symbols and lines on the graphs.

Figures 8 and 9: There are data series and their rolling-averages, but not trends.

Technical corrections

Lines 2 and 93: The word "seas" needs writing with a capital letter.

Lines 487 and 781: The word "Sea" needs writing in the plural.

Line 512: A mistype in the Persey Current name.

Figure 2: One mistype needs correcting: Jul should be replaced with Jan in the legend.

Figure 5: A mistype in the PINRO abbreviation.

---

## Editor Comment (EC1) · J. Hutchings (Editor) · 12 Jun 2018

Dear Dr. Leifer and co-authors,

We have received two reviews of your manuscript and based on these I can not recommend this paper is published. There are two fundamental issues in the interpretation your your results. Namely that the SST increase is not necessarily related to increased upwelling and that your calculation and consideration of albedo is not correct. In my reading of the manuscript I came to the same conclusion about the results related to changing sea ice conditions, and believe you have not considered the role of time in solar warming the upper ocean.

[Figure]

The reviewers have given advice on how to stream line the paper and make it suitable for publication. The methane analysis by itself may be of interest for publication. Given the inherent problems in the study in finding a link to sea ice or physical oceanography I do not believe The Cryosphere is a suitable journal for that study. You may reconsider your methodology, of course, and with new data or analysis methods be able to find linkages of the atmospheric methane to more distance ocean floor sources, and I would welcome a discussion on a paper documenting that in the future.

You are welcome to continue with the process of responding to reviews and revising your paper. Please do take into consideration the major nature of the criticism, however. At the stage I feel it is unlikely this study is publishable in it's present form, and it would be a very different paper should you address the key problems in your study.

Please do consider publishing the methane work, and I do hope that the reviews help you refine your methods to better understand the role of the ocean in the observed atmospheric hotspots. With very best regards, Jenny

---

## Author Comment (AC2) · 25 Jul 2018

Response to Review 2 Review of "Satellite ice extent, sea surface temperature, and atmospheric methane trends in the Barents and Kara Seas" by Ira Leifer et al.

General comments The submitted paper looks rather interesting. The authors carried out study in considering one of the current warming consequences, namely: increasing methane concentrations. They made a detailed enough review of the relevant literature and presented rather detailed analysis of the subject.

However, the paper needs reworking in some ways listed below.

[Figure]

> Thank you. Our primary study finding is that heat transfer by currents is driving increasing seabed methane emissions and that shoaling by the same currents is allowing this seabed methane to escape to the atmosphere. Methane shoaling is a newly described process that is neglected in a range of published studies to date, which focus on the local area; yet shoaling can occur far downcurrent, potentially in waters of another nation (Russia). We also provide novel SST climatology that shows that currents are not only important on the gross scale (WIGAR), and on the regional scale in parts of the Barents Sea, but even on the small scale – tens of kilometers. Specifically, if the SST trends only represented a skin effect, it would not have any effect on seabed methane.

The introduction and methods seem to be too long. These two chapters take about 11 pages of text compared to about 10 pages for the results and discussions.

> We have shortened significantly, rearranged and removed redundancy of the intro material, and moved information that is relevant but not central into supplemental material sections S1 and S2, for example. the Alaskan arctic methane review, and the summary of IASI satellite details and its validation (this supports the IASI Arctic review that remains in the main text). We deleted the discussion of SWIR satellite methane sensors (not relevant).

> Intro material is now 5.5 pages, Methods 1.25 pages, Results 6 pages, Discussion & Conclusion 7 pages.

Subchapter 2.3 Setting looks redundant, especially for the chapter describing materials and methods, except the very last paragraph.

> All of Subchapter 2.3 was moved into section 1.4 and 1.5 and duplication was removed. As well, these sections, and the beginning introduction were reorganized on a sentence by sentence basis to flow better and remove redundancy.

Some inaccurate using of Barents Sea currents names take place in the paper (see

below).

> We have worked hard to correct names, add additional names as needed to the maps, and improved clarity of the maps. We also added a few more details on current flow strength (Sv) and citations.

Drawing conclusions about variability in currents taking into account only variability of SST is incorrect.

> We recognize this and have improved the text. See extensive responses to Reviewer 1. That said, we argue that if SST trends only represented a skin effect or changes in downwelling thermal radiation, they would not have an effect on seabed methane. Moreover, the importance of shoaling argues that just as seabed methane is not being effectively transported vertically to the sea surface, then atmospheric heat transfer to the upper ocean cannot be effectively transported to the seabed (to affect seabed methane emissions).

Most figures are too small, of a poor quality and, as a result, hardly readable.

> The original uploaded figures are high quality. It is unclear why the system provided you with low quality images and we apologize.

Specific comments

Line 26: The Murman Coastal Current flows mainly eastward. The Murman Current also flows eastwards towards Novaya Zemlya, and in the eastern part, the flow (usually called as the Novaya Zemlya Current) is generally northeastwards along the slopes of the North Kanin, Goose and Novaya Zemlya Banks.

> Fixed in Fig. 4. Also added to the text in Section 2.3.1.

Line 27: SST depends on both currents and air temperature. Air temperature plays a key role in SST variability in shallow waters of the south-eastern Barents Sea and southern Kara Sea. Heat fluxes between ocean and atmosphere are important there.

> This is a highlight (and thus must be very short). Not really certain about this comment. We now note that currents transport heat.

Line 55 and on: There is one confusing term in the paper, namely: albedo. Usually, surface albedo is defined as the ratio of irradiance reflected to the irradiance received by a surface, and ranges from about 0.9 for fresh snow to about 0.04 for charcoal, one of the darkest substances. In the paper, albedo seems to be defined as the ratio of irradiance absorbed to the irradiance received, doesn't it?

> This was a mistake and is fixed.

Line 60: The word "Arctic" is unnecessary in the phrase "the East Siberian Arctic Sea".

> We think this is still an open question as there is quite a bit of literature that does refer to it as the ESAS (e.g., Shakhova et al., 2014)

Line 330: Novaya Zemlya is an archipelago, not an island, and includes two large islands (Southern and Northern) and lots of small.

> Fixed. Thanks. To emphasize, the Matochkin Strait is now labeled in Fig. 2.

Line 327: The phrase "The relatively shallow (230-m average depth) Barents Sea (Fig. 4) is characterized by a deep Arctic shelf. . ." is ununderstandable.

> Agreed. Rewritten to be understandable.

Line 335 and on: The Norwegian, Greenland, Barents, Kara and some other seas are included in the Arctic Ocean. Therefore, it is better to say the Arctic Basin (the central area of the Arctic Ocean beyond all these seas) in this context.

> Fixed everywhere including in the figure captions. Thanks,

Line 338: There is no current in the Barents Sea called "Bear Island Channel Current". There are warm North Cape current flowing eastwards in the Bear Island Trough and cold Bear Island Current flowing southwest along the southeastern slope of the

Spitsbergen Bank.

> We now note that the BICC designation is our own notation for the warm current, referenced in Li and McClimans (1998). The SST climatology clearly shows its impact on heat transport towards the Svalbard Bank, an area of shoaling. The change in PC to BIC occurs near Hopen and is noted and changed in the main figures; it was correctly shown in Supp. Fig. S1.

Line 340: The Grand bank mentioned in the paper is commonly known as the Great bank.

> Corrected here, and in figures 2 and 4 and in the supplemental material

Lines 385–387: There is a repetition in this place: "cooler surface water . . . flows in NZCC and exit through the Kara Strait". The NZCC abbreviation expansion needs being done at the first mention in line 385.

> This entire paragraph (379-394) on currents in the Kara Sea was confused and mixed up, jumping back and forth between topics. It has been rewritten for clarity and to remove duplication.

Line 403: The cold Bear Island Current flows there, not the Persey Current that is located further northeast.

> Thanks. Some of the maps we relied on to assemble our map did not designate the BIC, Corrected in figures and texts, and citations added.

Line 404: Both focus areas (A8 and A10) are only influenced by Atlantic waters, the Persey Current does not influence them (this is clearly seen in picture 4a).

> Not sure how we missed this. Thanks. The summary of focus areas A8-A10 rewritten for correctness and clarity.

Line 418: The Saint Anna Trough is a correct name.

> Thanks for identifying that typo. Corrected

Line 418: The focus areas are shown in figure 4, not 3.

> Corrected

Line 430: According to bathymetry shown in figure 4a, focus area 7 covers the shelf slope, and focus area 8 covers depths of more than 200 m. They do not seem to cover banks.

> Corrected. Thanks.

Lines 435 and 451: All three northern focus areas (A1–A3) are located south of the FJL, none of them are east of the FJL.

> Thanks, corrected to south and southeast. Also in line 463 and elsewhere. Also corrected is the north arrow in Fig. 4b.north arrow in Fig. 4b

Line 450: According to figure 7, trends are almost absent in some areas (A3 and A6, for example).

> Re-organized sentence to direct reader to Table 1, which shows that dSST/dt>0 for all focus areas.

Line 453: This is a moot point that trends in SST are consistent with strengthening or weakening of currents. Advection of waters with higher temperatures and local conditions affect these trends as well. For example, the volume flux into the Barents Sea was rather low from 2007 to 2015 (ICES. 2017. Report of the Working Group on the Integrated Assessments of the Barents Sea. WGIBAR 2017 Report 16-18 March 2017. Murmansk, Russia. ICES CM 2017/SSGIEA:04. 186 pp.).

> Agreed that this sentence is redundant and not needed. Deleted.

Lines 483–484: Measured volume fluxes through the Barents Sea Opening (see reference above) are weakly consistent with this statement.

> Sentence added to cite WGIBAR 2017

Lines 489–490: The mentioned importance is not clear from figure 9 at all.

> Should have referenced Fig. 10. Fixed.

Line 491: It is ununderstandable what current is mentioned (BIC). Along the eastern coastline of the Svalbard, the cold East Spitsbergen Current only flows southwards.

> We agree this was unclear – we revised this paragraph to only focus on the flow of warm water into the south and west and east Barents Sea (and the BICC spur). The relationship between currents and SST climatology is now elucidated much more clearly in Supp. Fig. S3, and provides new fine scale detail of the importance and how this changes with season. This paragraph now includes these details.

Lines 498–499: This is the Bear Island Current, not the Persey Current.

> Yes. Thanks. Rewritten.

"In June, the edge of the cold (Arctic water) Percey Current/Bear Island Current corresponds well with the warm water's edge and also corresponds fairly well with the median ice edge location. Southeast of Svalbard, the Bear Island Current penetrates southward as a narrow extension of cold water ending south of Bear Island. Slightly cooler water is observed over the two banks in the central Barents Sea."

Line 540: The Kara Sea is ice-free in September and almost ice-free in August but not in July (even during the last warm period). In July, the northern and northeastern parts of the Kara Sea as well as the area close to the eastern coastline of the Novaya Zemlya Archipelago are still ice-covered often.

> Yes – this is shown in Fig. 2a

Lines 614–617: The authors need to pay more attention to winds over the Barents and Kara Seas; the role of atmospheric circulation seems to be underestimated.

[Figure]

> It is hard to explain the methane anomalies by distant sources from Europe or Russia for two reasons, described in the paragraph. The paragraph was rewritten for clarity:

> We have only considered the effect of vertical mixing due to wind. At these scales, wind transport shoves water to the right (Ekman transport), which is beyond the scope of our study. In an earlier paper (McClimans & Nilsen, 1993) it was shown that most details of the currents, including the Polar Front were produced by controlling the inflows of Atlantic and Arctic Surface waters from the NAC and the PC. [McClimans, T. A. and Nilsen, J. H, Laboratory simulation of the ocean currents in the Barents Sea. Dynamics of Atmospheres and Oceans. 19:3-26].

"The presence of a localized, atmospheric CH4 anomalies can reflect either local seabed emissions being vertically transported to the WML / atmosphere, or distant seabed emissions that currents transport laterally upslope into the WML where air sea exchange transports the CH4 into the atmosphere. Detangling these processes leverages the strength of the continuous and synoptic view of satellite data. Atmospheric transport of a distant source would not be localized."

Line 682: The evidence of the statement that currents are strengthening is not provided. The phrase "larger oceanographic trends" is ununderstandable.

> As written, it is simply one of several possibilities (or hypotheses). We have rewritten the paragraph (deleting the phrase "larger oceanographic trends," which is beyond the scope of this study – whether the driving force is the NAO, or global warming, or other processes). In summary, we simply conclude that greater heat transport is critical for increased tropospheric methane and is consistent with the hypothesis that it is strengthening heat transport by currents to both the seabed and sea surface.

"There are a number of possible hypotheses for why SST is warming fastest in regions along the Murman Current and NAC. One is sea-ice retreat; however, the warming occurs several months after the retreat of the sea ice. Another is that the pycnocline is becoming shallower, allowing more cooling to the atmosphere. This would imply a

weakening of storms and winds – which firstly is inconsistent with warmer SST, and secondly, there is no indication that Barents Sea storminess is changing or progressing further northwards (Koyama et al., 2017). Another hypothesis is that increasing current transport of heat is driving the SST warming. Although SST derives from several factors including heat transfer from the bulk ocean (i.e., currents), its co-spatial relationship to enhanced CH4 anomaly is consistent with currents playing a major role both at the sea surface (SST anomaly trend) and at the seabed. Greater heat transport could occur from strengthening currents, or warming currents, or a combination of both. "

Lines 731 and 732: There is some mess in using "northwest" and "northeast". The FJL and St. Anna Trough are in the northeastern Barents Sea, not in the northwestern. The southwestern and southern parts of the Barents Sea are ice-free year round, but not the northeastern part.

> Thanks for catching that mistake. Fixed.

Figure 4a: The East Greenland Current looks like having the wrong direction. This current is flowing southwards (Skjoldal H.R. (Ed.). 2004. The Norwegian Sea Ecosystem. Tapir Academic Press, Trondheim. 559 pp.; Dickson R.R., Meincke J., Rhines P. (Eds). 2008. Arctic–Subarctic Ocean Fluxes: Defining the Role of the Northern Seas in Climate. Springer, Dordrecht. 736 pp.). The detailed description of the Barents Sea currents is also presented in: Jakobsen T., Ozhigin V.K. (Eds). 2011. The Barents Sea: ecosystem, resources, management: Half a century of Russian-Norwegian cooperation. Tapir Academic Press, Trondheim. 825 pp. (scanned version of this book is available at https://brage.bibsys.no/xmlui/handle/11250/109444).

> Fixed. This was an error of the drawing program that we didn't catch. We also improved the currents in Fig. 4a for the southeast Barents Sea, and cite Jakobsen and Ozhigin (2011).

Figure 6: The caption mentions figure 3c that is absent.

> Fixed – changed to 4a

Figures 7, 8 and 9: There is no explanation for symbols and lines on the graphs.

> Added to caption that lines are three year rolling average, and symbols are not.

Figures 8 and 9: There are data series and their rolling-averages, but not trends.

> Figures 8 and 9 captions corrected.

Technical corrections

Lines 2 and 93: The word "seas" needs writing with a capital letter.
 Lines 487 and 781: The word "Sea" needs writing in the plural.

> Corrected throughout the entire manuscript

Line 512: A mistype in the Persey Current name.

> Some literature refers to the Percey Current, some to Persey Current. We now note this.

Figure 2: One mistype needs correcting: Jul should be replaced with Jan in the legend.

> Yes! Thanks. Fixed.

Figure 5: A mistype in the PINRO abbreviation.

> Typo fixed. Thanks
* * *
**a)**
BICC - Bear Island Channel Current
BIC - Bear Island Current
ESC - East Spitsbergen Current
MC - Murman Current
MCC - Murman Coastal Current
NCC - Norwegian Coastal Current
NAC - Norwegian Atlantic Current
NZC - Novaya Zemlya Current
NZCC- Novaya Zemlya Coastal Current
PC - Percey Current
SCC - Spitsbergen Coastal Current
WSC - West Spitsbergen Current

*BI* – Bear Island
*CB* – Central Bank
*FJL*–Franz Josef Land
*GB* – Great Bank
*KS* – Kola Section
*NZ* – Novaya Zemlya
*NZB* –Novaya Zemlya Bank
*SAT*– St. Anna Trough
*SB* – Svalbard Bank
*YP* – Yermak Plateau

**b)**

1–Arctic Franz Josef Land North (AFN)   6–Svalbard, West, S. West (SWSW)
2–Arctic Franz Josef Land South (AFS)   7–Svalbard, South (SS)
3–Svalbard North East (SNE)             8–Murman, North (MN)
4–Svalbard West, North (SWN)            9–Novaya Zemlya, S. West (NZSW)
5–Svalbard West (SW)                     10–Barents Sea, center (BSC)

**Fig. 1.** Revised figure 4

---

## Author Comment (AC3) · 25 Jul 2018

Dear Jenny,

Although upon submission, I felt that this manuscript was one of the best edited I have submitted, I find myself agreeing that the manuscript needed major work at the editorial level along. In that regards, we have followed the suggestion of Reviewer #2 and greatly reduced (about 40%) the introductory material both through moving some material to supplemental and by removing repetition after significant re-organization to improve the flow.

[Figure]

With respect to the very significant point that SST is not current, we acknowledge that this is true, but missed our point, which we evidently did not explain clearly. Specifically, that fact that there is CH4 emissions requires seabed heating, which cannot be explained if SST trends only represent a skin effect or changes in downwelling thermal radiation. Moreover, the importance of shoaling argues that just as seabed methane is not being effectively transported vertically to the sea surface, then atmospheric heat transfer to the upper ocean cannot be effectively transported to the seabed (to affect seabed methane emissions). That said, we recognize that correlation is not causation, and that proving causation is beyond the scope of this manuscript. Thus, we now reference the hypothesis that SST trends are consistent with increasing heat transport by currents.

We do acknowledge that SST depends on a number of factors including downwelling radiation and atmospheric temperature (and hence transport). Thus although our analysis relied on the spatial localization of SST changes that would be hard to explain from large scale atmospheric processes, we agree that this must be introduced and properly discussed.

We also identified many small errors in Fig. 4a, which we have corrected. Currents in Fig. 4a also have been improved. Also, we have improved all other figures for clarity and readability and as suggested by the reviewers.

We would like to argue that the key highlights from our manuscript make it still suitable for The Cryosphere, the novel mechanisms of shoaling which affects our understanding of the fate of seabed methane, particularly in a sea like the Barents Sea, where currents drive water to shoal as it exits and crosses banks. Also important is the discovery of methane sources associated with Franz Josef Land and off the west coast of Novaya Zemlya and from the areas of predicted oil and gas reservoirs in the Barents Sea.

Although we describe the relationship of SST anomaly trends to currents as a hypothesis, we believe the synoptic satellite data that our analysis leverages is sufficiently

strong and consistent with the hypothesis that it would be of value to The Cryosphere community to spark discussion and to encourage the collection of field data to validate or disprove the hypothesis.

Thus we respectfully request your consideration of our paper to continue in the review process and/or as a new submission.

Sincerely, Ira Leifer and co-authors

––––––––––––––––––––––––––––